# Plasticity from Structured Sparsity: Mastering Continual Reinforcement Learning through Fine-Grained Network Allocation and Dormant Neuron Exploration

## Abstract

Continual reinforcement learning faces a central challenge in striking a balance between *plasticity* and *stability* to mitigate catastrophic forgetting. In this paper, we introduce **SSDE**, a novel structure-based method that aims to improve plasticity through a fine-grained allocation strategy with **S**tructured **S**parsity and **D**ormant-guided **E**xploration. Specifically, SSDE decomposes the parameter space for each task into *forward-transfer* (frozen) parameters and *task-specific* (trainable) parameters. Crucially, these parameters are allocated by an efficient co-allocation scheme under sparse coding, ensuring sufficient trainable capacity for new tasks while promoting efficient forward transfer through frozen parameters. Furthermore, structure-based methods often suffer from rigidity due to the accumulation of non-trainable parameters, hindering exploration. To overcome this, we propose a novel exploration technique based on sensitivity-guided dormant neurons, which systematically identifies and resets insensitive parameters. Our comprehensive experiments demonstrate that SSDE outperforms current state-of-the-art methods and achieves a superior success rate of 95% on the CW10-v1 Continual World benchmark.

## 1 Introduction

While human beings demonstrate remarkable abilities to adapt knowledge from previous tasks to new challenges without forgetting, AI models, particularly reinforcement learning (RL) agents, struggle in non-stationary environments (Thrun, 1998; Choi et al., 1999). Research in continual RL aims to overcome this by enabling agents to learn sequential tasks (Wołczyk et al., 2021; Khetarpal et al., 2022). However, this work faces a major challenge: catastrophic forgetting (McCloskey & Cohen, 1989; Caruana, 1997). This problem comes from the difficulty in balancing *plasticity* and *stability* in learning systems. *Plasticity* allows agents to quickly adapt to new tasks, while *stability* ensures that previously learned skills are retained (Abbas et al., 2023; Dohare et al., 2024). Existing works have pursued three main approaches: (i) *rehearsal*-based, (ii) *regularization*-based, and (iii) *structure*-based (Khetarpal et al., 2022; Wang et al., 2024). Notably, *rehearsal*-based and *regularization*-based methods offer relatively limited control over *stability*, as experience replay and constrained learning pose the risk of unintended interference with previously learned parameters. In contrast, *structure*-based methods excel at preserving *stability* by explicitly forming task boundaries and allocating task-specific sub-networks, effectively minimizing interference and preventing catastrophic forgetting.

Structure-based methods often leverage sparsity to accommodate the sub-networks for multiple tasks within a shared parameter space (Wang et al., 2022; 2024). *PackNet* (Mallya & Lazebnik, 2018) prunes parameters after each task, retaining only the most crucial ones. *CoTASP* (Yang et al., 2023) generates sparse binary masks based on task descriptions to calibrate the output of each layer. These masks, initialized through sparse coding and dictionary learning, are updated via gradient computed from RL objectives. However, treating sub-network allocation as a unified process leads to a gradual reduction in trainable parameters as more tasks are introduced.

This limits the model's capacity to adapt and compromise *plasticity*, especially for complex tasks (scaling law (Hilton et al., 2023)). Moreover, these methods demand substantial computational overhead for sub-network allocation due to resource-intensive operations like pruning or gradient-based optimization. To enhance the *plasticity* of structure-based methods, it is essential to allocate sufficient trainable parameters for new tasks while effectively utilizing trained parameters from previous tasks during inference. Both are critical for maintaining the expressiveness of the sub-network policy and improving the continual learning performance of structure-based methods.

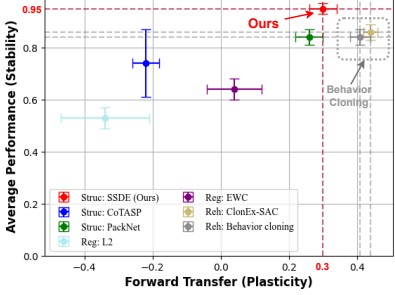

Figure 1: *Plasticity-stability* trade-off on CW10-v1. **SSDE** achieves SOTA *stability* of 95%. *Plasticity* is the normalized step to learn a task, where **SSDE** is competitive to BC baselines that benefit from replay.

In this paper, we present **SSDE**, a novel method for "enhancing *plasticity* through **S**tructured **S**parsity and **D**ormant neuron-guided **E**xploration", designed to optimize the three core aspects of continual RL policies: **(i) Allocation**: SSDE introduces a **fine-grained co-allocation** strategy based on sparse coding, which explicitly decomposes sub-network parameters into forward-transfer (fixed) and task-specific (trainable) components, and ensures sufficient capacity for learning new skills while maintaining knowledge transfer efficiency. **(ii) Inference**: SSDE incorporates a dedicated inference function with a novel **trade-off parameter** that dynamically balances forward-transfer and task-specific parameters, preventing frozen parameters from overshadowing trainable ones and expanding the solution space for flexible and diverse inference strategies. **(iii) Training**: SSDE introduces a *sensitivity-guided dormant* **neuron** algorithm to enhance expressiveness of sparse policy which restrictive capacity for trainable neurons. By identify neurons unresponsive to input sensitivity, it addresses the unique expressivity challenges of sparse sub-networks.

Together, the strategic combination of fine-grained co-allocation and exploration with *dormant* neurons establish a robust foundation for SSDE to significantly enhance the *plasticity-stability* trade-off. We show SSDE not only achieves SOTA *stability* but also achieves competitive *plasticity* even when compared to strong behavior cloning baselines that benefit from data replay (Figure 1). We also show the consistency of SSDE's performance across both v1 & v2 of Continual World benchmark (Table 3 & Table 2). A case study on co-allocated masks with structured sparsity highlights that our approach significantly improves parameter utilization while drastically reducing allocation time (Table 1 & Figure 4). Visualizations of sub-network masks further demonstrate that structural sparsity effectively captures task similarities (Figure 5 & Figure 13). Finally, a comprehensive ablation study (Table 4) confirms that SSDE's core components are crucial for driving the success of the model.

## 2 RELATED WORKS

Continual RL, a.k.a. lifelong RL, seeks to develop agents capable of continuously learning from a sequence of tasks without forgetting previous knowledge. For a comprehensive survey, we refer readers to (Khetarpal et al., 2022), and for a formal definition of continual RL agents, see (Abel et al., 2023). A detailed illustration of rehearsal-based, regularization-based, and structure-based strategies is provided in Appendix A.3. Among these approaches, the SOTA for continual learning on Meta-World manipulation tasks (Yu et al., 2019) is held by the rehearsal-based method CloneEX-SAC (Wolczyk et al., 2022). By storing previous task data and policies for behavior cloning, it achieves high forward transfer via intensive data replay, though at a significant computational cost. In contrast, structure-based methods avoid data reuse and use a single set of parameters to represent multiple policies, enabling more efficient training and inference.

While structure-based methods reduce task interference by using sub-networks, they often pursue sparsity-driven allocation, which sacrifices capacity and hinders adaptation (*plasticity*). Pack-Net (Mallya & Lazebnik, 2018) prunes the network after each task, fine-tuning the dense policy into a sparse one by retaining the most important parameters, albeit with significant computational effort. HAT (Serrà et al., 2018) learns a hard attention mask for each task by adding a small number of trainable weights that are updated alongside the main model. CSP (Gaya et al., 2023) progressively expands the subspace of policies by integrating new policies into the space as anchors, if learning with the new parameters brings positive performance gain. Rewire (Sun & Mu, 2023) employs a dif-

ferentiable wiring mechanism to adaptively permute neuron connections, enhancing policy diversity and stability in non-stationary environments. Recently, sparse prompting-based approaches have emerged, effectively bridging cross-modality task relationships with parameter allocation strategies. TaDeLL (Rostami et al., 2020) employs a coupled dictionary optimization to augment task descriptors and policy parameters, initializing the policy for a new task as a sparse linear combination over a shared basis. CoTASP (Yang et al., 2023) extends this by initializing sparse sub-network masks through sparse encoding and dictionary learning, which are updated during RL via gradient optimization. Though both works focus on leveraging task similarities for parameter allocation, they overlook a critical issue of sparse networks progressively losing trainable parameter capacity, which hinders the acquisition of new skills. Our work overcomes this limitation with a novel co-allocation strategy built on sparse coding, designed to ensure effective forward-transfer parameter allocation while simultaneously dedicating sufficient capacity for trainable parameters. SSDE further achieves fully preemptive allocation, removing the need for computationally intensive dictionary learning or iterative updates used in CoTASP, significantly enhancing allocation efficiency.

Our work further bridges continual RL with the recently proposed *dormant neuron phenomenon* (Sokar et al., 2023) to address a key question: *How can structure-based continual RL agents use their sparse sub-networks to their full potential?* Sokar et al. (2023) proposes *ReDo*, a mechanism that periodically resets inactive neurons from full-scale dense policies to restore network capacity without significantly altering policy. It identifies dormants using a simple yet effective method based on neuron activation scales. In context of structure-based continual RL, expressivity challenge is more pronounced due to sparse sub-networks, where a substantial portion of parameters are frozen, leaving only a small fraction trainable. SSDE extends the dormant neuron concept by proposing a sensitivity-guided dormant that intuitively identifies neurons unresponsive to observation changes, enhancing the sparse policy's responsiveness to crucial states. Integrating this phenomenon into continual RL is crucial, as expressivity is directly tied to *plasticity*, especially in sparse sub-networks where limited capacity hinders adaptability. To the best of our knowledge, SSDE is the first work to address expressivity limitations of sparse policy networks in continual RL.

## 3 PROBLEM FORMULATION

Our work focuses on solving continual RL problems under a task-incremental setting, following (Wołczyk et al., 2021; Yang et al., 2023). Formally, we aim to train a single RL policy to solve a sequence of N distinct tasks $\mathcal{T}_{cl} = \{\mathcal{T}_1, ..., \mathcal{T}_N\}$, where each task $\mathcal{T}_k$ is observed sequentially and defined as a Markov Decision Process (MDP), $\mathcal{T}_k = \langle \mathcal{S}^{(k)}, \mathcal{A}^{(k)}, p^{(k)}, \mathcal{R}^{(k)}, \gamma \rangle$. Here, $\mathcal{S}^{(k)}$ represents the state space, $\mathcal{A}^{(k)}$ is the action space, $p^{(k)} : \mathcal{S}_t^{(k)} \times \mathcal{A}_t^{(k)} \to \mathcal{S}_{t+1}^{(k)}$ is the transition probability function, $\mathcal{R}^{(k)} : \mathcal{S}_t^{(k)} \times \mathcal{A}_t^{(k)} \to \mathbb{R}$ is a reward function, and $\gamma \in [0, 1)$ is the discount factor. The goal of the agent is to learn an optimal continual RL policy $\pi_\theta^*$ that performs well on the entire distribution of tasks through sequential task interaction,

$$\theta^* = \arg\max_\theta \sum_{k=1}^{\mathcal{T}} \mathbb{E}_{\pi_\theta} \left[ \sum_{t=0}^{\infty} \gamma^t \mathcal{R}^{(k)}\left(s_t^{(k)}, a_t^{(k)}\right) \right]. \tag{1}$$

Structure-based continual RL tackles the challenge of *plasticity-stability* trade-off with a strategy of dynamically partitioning the policy network into task-specific sub-networks, minimizing task interference and preserving the degradation of earlier behaviors. Formally, for each task $\mathcal{T}_k$, it establishes a mapping $\phi : \theta \times \text{task}_{id}(k) \to \theta_k$ that automatically maps the network parameters $\theta$ and task identity $\text{task}_{id}$ to generate a dedicated sub-network policy $\pi_{\theta_k}$, where $\theta_k \subseteq \theta$. Crucially, the space for the sub-network parameters can be decomposed into two parts: $\theta_k = \{\theta_k^{fw}, \theta_k^{sp}\}$, where $\theta_k^{fw}$ represents a group of the *forward-transfer* parameters shared with previous tasks $\theta_1, ..., \theta_{k-1}$, frozen after training and used for inference only. Formally, $\theta_k^{fw} = \left( \bigcup_{i=1}^{k-1} \theta_i \right) \bigcap \theta_k$, and the remaining are *task-specific* parameters, updated solely for learning the current task, i.e., $\theta_k^{sp} = \theta_k \setminus \theta_k^{fw}$. Note that each task only updates its *task-specific* parameters $\theta_k^{sp}$, while the *forward-transfer* parameters $\theta_k^{fw}$ remain fixed to prevent task interference. To scale the agent's capability for handling multiple

tasks, the sub-network parameters are typically sparse. For efficient task allocation, we employ a neuron-level partitioning method to establish task boundaries. For each layer $l$, sub-networks are prompted by applying binary masks $\phi_k^{(l)}$ to the outputs of the $l$-th layer $y^{(l)}$, generating calibrated network outputs as follows:

$$\boldsymbol{y}_k^{(l)} = \phi_k^{(l)} \otimes f(\boldsymbol{y}_k^{(l-1)}; \theta_k^{(l)}), \tag{2}$$

where $f(\cdot)$ is the conventional inference function for the $l$-th layer, $\theta_k^{(l)} = \{\theta_k^{fw_{(l)}}, \theta_k^{sp_{(l)}}\}$, and $\otimes$ is element-wise multiplication. The key to the structure-based approach lies in determining how to allocate $\theta_k^{fw}$ and $\theta_k^{sp}$ for each task to maximize the use of learned knowledge through $\theta_k^{fw}$ while ensuring sufficient capacity in $\theta_k^{sp}$ to capture new skills. However, existing structure-based RL methods overlook these fine-grained relationships and address the allocation of $\theta_k$ as a unified process.

## 4 METHODOLOGY

In this section, we propose **SSDE** (*plasticity* through **S**tructured **S**parsity with **D**ormant neuron-guided **E**xploration). In Sec 4.1, we introduce a **fine-grained co-allocation** algorithm that allocates parameters for each task regarding task global relationship and local structure. The allocated parameters are decomposed into *forward-transfer* (fixed) and *task-specific* (trainable), allocated under an objective of preserving knowledge from previous tasks for forward transfer while maximizing the number of trainable parameters for increased network plasticity (Sec 4.1.1). Then we detail a sub-network masking mechanism that facilitates task-specific prompting during inference and training. Using neuron-level and parameter-level masks, sub-networks can be efficiently frozen or activated as needed (Sec 4.1.2). In Sec 4.2, we introduce a **dormant neuron-guided exploration** strategy that re-activates the sensitivity-guided dormant neurons to overcome the expressiveness challenges of training sparse sub-network policies.

### 4.1 FINE-GRAINED SUB-NETWORK ALLOCATION

Sparse prompting-based approaches (Yang et al., 2023; Reimers & Gurevych, 2019) enhance plasticity by assigning sparse codes to respective tasks, which are then transformed into neuron masks to generate dedicated sub-networks. Building on this foundation, we propose a **sub-network co-allocation strategy** that leverages *global* task correlation and task-specific *local* dictionaries for effective allocation. Specifically, we introduce **global** task-related sparse prompting, denoted as $\alpha_{[\Gamma]}$, derived from embeddings of text descriptions encoded by pre-trained Sentence-BERT Reimers & Gurevych (2019) and a global coding dictionary, to capture shared task relationships. Crucially, we also introduce **local** task-capacity sparse prompting, $\alpha_{[\Lambda]}$, derived from individual local dictionaries, to ensure sufficient capacity for task-specific parameters. These two components synergistically co-allocate dedicated forward-transfer and task-specific parameters in sparse sub-networks, enhancing network plasticity. During reinforcement learning, we incorporate a **fine-grained masking mechanism** to efficiently manage the *forward-transfer* parameters, freezing them for stability and forward transfer, while selectively updating *task-specific* parameters to integrate new knowledge.

#### 4.1.1 CO-ALLOCATION WITH SPARSE PROMPTING

To obtain a global task-related sparse prompting $\alpha_{k[\Gamma]}$, we begin by generating task embeddings $e_k \in \mathbb{R}^m$ of tasks $\mathcal{T}_k$ by encoding their corresponding task textual descriptions using a pre-trained LM. For each layer-$l$ in the sub-network, we construct a shared space among all the tasks as an over-complete dictionary $\boldsymbol{D}^{(l)} \in \mathbb{R}^{m \times n^{(l)}}$, where $n^{(l)}$ is the number of neurons at layer-$l$ in the full network. Elements in $\boldsymbol{D}^{(l)}$ are sampled from normal distribution $\mathcal{N}(0, 1)$ and $\boldsymbol{D}^{(l)}$ is fixed for all task in $\mathcal{T}_{cl}$. We aim to learn a sparse prompting $\alpha_{k[\Gamma]}^{(l)} \in \mathbb{R}^{n^{(l)}}$ that could reconstruct the task embeddings $e_k$ as a linear combination of neuron's representations, i.e., atoms from the dictionary. The sparse prompting $\alpha_{k[\Gamma]}^{(l)}$ can be obtained by optimizing the Lasso problem formalized as follows,

$$\alpha_{k[\Gamma]}^{(l)} = \underset{\alpha_{k[\Gamma]} \in \mathbb{R}^m}{\arg\min} \frac{1}{2} \|e_k - \mathbf{D}^{(l)} \alpha_{k[\Gamma]}^{(l)}\|_2^2 + \lambda_{[\Gamma]} \|\alpha_{k[\Gamma]}^{(l)}\|_1, \quad \text{for layer } l = 1, \ldots, L-1, \tag{3}$$

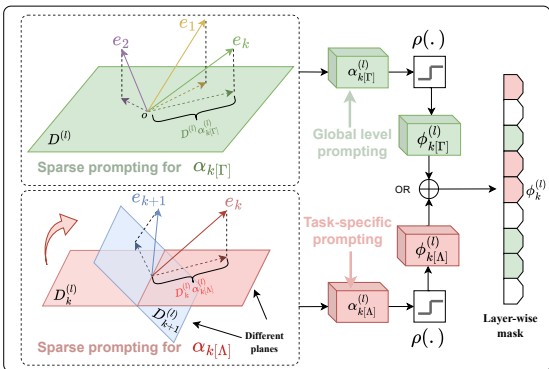

Figure 2: **Co-allocation with sparse prompting** aims to learn two sets of calibration embeddings, $\boldsymbol{\alpha_{k[\Gamma]}}$ and $\boldsymbol{\alpha_{k[\Lambda]}}$, which generate neuron-level binary calibration masks $\boldsymbol{\phi_k^{(l)}}$ to determine the sub-net structure for the $l$-th layer. **Upper**: a global-level sparse coding process learns $\boldsymbol{\alpha_{k[\Gamma]}}$ by projecting different task embeddings onto a shared plane of $\boldsymbol{D}^{(l)}$, assigning similar masks to similar tasks. **Lower**: a task-specific prompting process leverages random projection planes to learn $\boldsymbol{\alpha_{k[\Lambda]}}$, to increase the capacity for trainable parameters. Together, these processes co-allocate binary masks, promoting enhanced *plasticity*.

where $\|\cdot\|_p$ is the $L_p$ norm, $\lambda_{[\Gamma]}$ is a hyperparameter controlling the sparsity of the forward-transfer prompting $\alpha_{k[\Gamma]}^{(l)}$, and $L$ is the number of layers. A step function $\rho(\cdot)$ transforms the sparse prompting into a binary mask, i.e., $\phi_{k[\Gamma]}^{(l)} = \rho(\alpha_{k[\Gamma]}^{(l)})$. The binary mask $\phi_{k[\Gamma]}^{(l)} \in \{0,1\}^{n^{(l)}}$ selects the sub-network neurons for task $\mathcal{T}_k$ at layer-$l$.

Through this optimization, similar tasks will result in neuron masks that allocate similar neurons. However, this similarity introduces a challenge: the overlap of fixed neurons with previous tasks leads to fewer trainable parameters for new tasks, potentially limiting the network's capacity to learn and adapt. To fully leverage the limited training capacity, we implement a strategy to maximize separation between task representations. The goal is to reduce the overlap in neuron usage across different tasks. This is achieved by introducing a novel mechanism: the embedding for each task, $\mathcal{T}_k$, is projected using its own **unique** dictionary, $\mathbf{D}_k^{(l)}$. This approach allows for more distinct representations of each task, even when the original task descriptions are similar.

Specifically, each element in $\boldsymbol{D}_k^{(l)}$ is drawn from $\mathcal{N}(0,1)$. The task-specific prompting $\alpha_{k[\Lambda]}^{(l)}$ that selects the trainable neurons is learned by solving the following objective:

$$\alpha_{k[\Lambda]}^{(l)} = \arg\min_{\alpha_{k[\Lambda]} \in \mathbb{R}^m} \frac{1}{2}\|e_k - \mathbf{D}_k^{(l)}\alpha_{k[\Lambda]}^{(l)}\|_2^2 + \lambda_{[\Lambda]}\|\alpha_{k[\Lambda]}^{(l)}\|_1, \quad \text{for layer } l = 1, \ldots, L-1, \quad (4)$$

where $\lambda_{[\Lambda]}$ is a hyperparameter to control the sparsity for $\alpha_{k[\Lambda]}^{(l)}$. To efficiently compute the sparse promptings, we employ a Cholesky-based implementation of the LARS algorithm (Efron et al., 2004), which offers a good balance of performance and ease of implementation. The binary mask for random projection, $\phi_{k[\Lambda]}^{(l)} = \rho(\alpha_{k[\Lambda]}^{(l)})$, is obtained by applying a threshold function $\rho(\cdot)$ to $\alpha_{k[\Lambda]}^{(l)}$. The final fine-grained sub-task masks, $\phi_k^{(l)}$, are derived by combining the two groups of masks: $\phi_k^{(l)} = \phi_{k[\Gamma]}^{(l)} \vee \phi_{k[\Lambda]}^{(l)}$, where $\vee$ represents the element-wise OR operation, and each element in the mask is a Boolean value (0 or 1).

Co-allocated final masks for each task's sub-network would convey high-quality forward-transfer parameters for fast adaptation, meanwhile also providing sufficient trainable parameters for current task updates. This mask learning process is computationally and data-efficient by using only the task description embeddings without dependence on any gradient optimization.

### 4.1.2 Fine-grained Sub-network Masking

With the neuron level mask $\phi_k^{(l)}$, we can investigate which parameters in weight matrix $W^{(l)} \in \mathbb{R}^{n^{(l)} \times n^{(l-1)}}$ are in used, i.e. the sub-network allocated, in Task $\mathcal{T}_k$. We denote a binary mask matrix by $\tilde{\Psi}_k^{(l)} \in \mathbb{R}^{n^{(l)} \times n^{(l-1)}}$, which can be computed by matrix-multiplication with $\phi_k^{(l)}$ and previous layer neuron mask $\phi_k^{(l-1)}$:

$$\tilde{\Psi}_k^{(l)} = \phi_k^{(l)}(\phi_k^{(l-1)})^T, \quad (5)$$

where the value 1 indicates the parameter is activated in task $\mathcal{T}_k$. Specially, $\phi_k^{(0)} = \mathbf{1}$ and $\phi_k^{(L)} = \mathbf{1}$ are vectors where all elements are ones. The element at row $p$ and column $q$ in matrix $\tilde{\Psi}_k^{(l)}$ is one

if and only if the $p$-th element of $\phi_k^{(l)}$ and $q$-th element of $\phi_k^{(l-1)}$ are both ones. Therefore, matrix $\tilde{\Psi}_k^{(l)}$ is more or equal sparse compared to $\phi_k^{(l)}$ and $\phi_k^{(l-1)}$. As mentioned in Section 2, the parameters indicated by $\tilde{\Psi}_k^{(l)}$ are allocated as the **sub-network parameters** for the current task $\mathcal{T}_k$ and as part of *forward-transfer* fixed parameters $\theta^{fw}$ in future tasks $\mathcal{T}_{k+1:N}$ to prevent catastrophic forgetting.

However, recent structure-based methods (Mallya & Lazebnik, 2018; Yang et al., 2023) freeze parameters in neuron level, in which **whole rows** of parameter in $W^{(l)}$ are fixed after training task $\mathcal{T}_k$. Many parameters, which are selected by $\phi_k^{(l)}$ but not covered by $\phi_k^{(l-1)}$, are **wasted**, as they remain scratch till the end of training. This strategy dramatically reduces the network plasticity and trainable parameters, leading to less network capacity for future tasks.

In order to solve this drawback, our work proposes **fine-grained** sub-network masking mechanism that freezes the **exact parameters** which have been trained in previous task $\mathcal{T}_{1:k-1}$. We maintain another mask matrix $\Psi_{k-1}^{(l)}$ for the frozen parameters, formally defined as performing element-wise OR operation across all seen fine-grained mask $\tilde{\Psi}_i^{(l)}$ for $i \leq k-1$, i.e. $\Psi_{k-1}^{(l)} = \vee_{i=1}^{k-1} \tilde{\Psi}_i^{(l)}$.

The fine-grained mask $\Psi_{k-1}^{(l)}$ covers the exact parameter that are used in tasks $\mathcal{T}_{1:k-1}$ and should be fixed starting from task $\mathcal{T}_k$. The forward function of layer-$l$ can be decomposed into two part regarding $\Psi_{k-1}^{(l)}$, inferring with the *task-specific* (trainable) parameters, and the *forward-transfer* (frozen) parameters, respectively.

$$\hat{\boldsymbol{y}}_k^{(l)} = \underbrace{\left((1 - \Psi_{k-1}^{(l)}) \otimes \tilde{\Psi}_k^{(l)} \otimes \boldsymbol{W}^{(l)}\right)}_{\text{task-specific parameters (trainable)}} \boldsymbol{y}_k^{(l-1)} + \underbrace{\beta}_{\text{trade-off}} \cdot \underbrace{\left(\Psi_{k-1}^{(l)} \otimes \tilde{\Psi}_k^{(l)} \otimes W^{(l)}\right)}_{\text{forward-transfer parameters (frozen)}} \boldsymbol{y}_k^{(l-1)} + b_k^{(l)} \otimes \phi_k^{(l)},$$

(6)

where $\hat{\boldsymbol{y}}_k^{(l)}$ is the pre-activation output, and the layer output is $\boldsymbol{y}_k^{(l)} = h(\hat{\boldsymbol{y}}_k^{(l)})$, with $h(\cdot)$ being the activation function. We introduce a trade-off parameter $\beta$, which plays a crucial role in achieving fine-grained control over the impact of forward-transfer parameters. As the task distribution evolves, the capacity of frozen parameters increases while the availability of trainable parameters decreases. Our fine-grained inference method utilizes $\beta$ to control the balance of pre-trained knowledge with the acquisition of new skills, preventing the pre-trained knowledge from overshadowing the trainable parameters, enhancing the plasticity, and enabling the model learn new tasks effectively. Figure 3 shows an example of the fine-grained inference procedure in SSDE.

**Learning** To optimize our proposed fine-grained sub-network allocation method, we update only the task-specific parameters using masked gradient descent as follows:

$$\theta^{(l)} \leftarrow \theta^{(l)} - \alpha \left(1 - \Psi_{k-1}^{(l)}\right) \otimes \boldsymbol{g}_k^{(l)},$$

(7)

where $\alpha$ is the learning rate and $\boldsymbol{g}_k^{(l)}$ is the gradient w.r.t. layer-$i$ parameters $\theta^{(l)}$, which are set to zeros when $\Psi_{k-1}^{(l)}$ is 1. In other words, we stop gradient for the term $\Psi_{k-1}^{(l)} \otimes \tilde{\Psi}_k^{(l)} \otimes W^{(l)}$ w.r.t. $W^{(l)}$. A detailed version of our proposed co-allocation method in **SSDE** is presented in Algorithm 1.

## 4.2 STRUCTURAL EXPLORATION WITH SENSITIVITY-GUIDED DORMANT NEURONS

Training sparse prompted sub-network policies in the continual RL domain often encounters a crucial challenge of limited expressivity due to the increasing *rigidity* of the policy network over tasks. As training progresses, the proportion of non-trainable parameters grows, dominating the network's output and restricting its adaptability to new tasks. This rigidity arises from the need to fix parameters to prevent catastrophic forgetting, reduces the availability of trainable parameters to adequately shape the policy for new learning (lose of plasticity). Consequently, only a subset of neurons remains active, leading to a less stochastic policy that becomes increasingly non expressive.

To enhance the adaptability of sparse sub-networks, we propose a novel **sensitivity-guided structural exploration** strategy facilitated by a newly defined **sensitivity dormant scores**. Motivated by dormant neurons phenomenon (Sokar et al., 2023), our approach involves **periodically resetting**

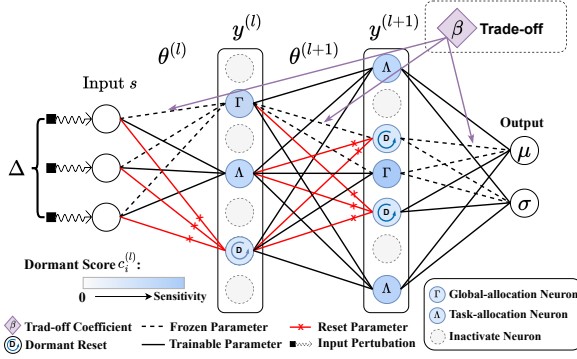

Figure 3: Illustration of **structural exploration with dormant neurons** in SSDE. Structural sparsity is achieved by generating a sub-network from neurons **co-allocated by two sparse prompting** processes ($\Gamma$ and $\Lambda$). Fine-grained inference is performed on it, with the **trade-off coefficient $\beta$** controlling the balance of *trainable* and *frozen* parameters. For structural exploration, the input of the sparse network is perturbed to maximize the sensitivity of active neurons. Neurons colored blue are evaluated on sensitivity score $c_i^{(l)}$, and inactive ones, denoted as ***dormant*** (marked 'D' with reset) are reset to enhance expressiveness.

neurons that have become unresponsive. Unlike prior work, which evaluates responsiveness solely by neuron activation scale, SSDE addresses the unique rigidity of sparse sub-networks, where limited trainable parameters hinder exploration and learning, often rendering the policy unresponsive to input variations (as highlighted in Appendix A.4.1).

To tackle this, we introduce *sensitivity-guided dormant neurons*, bridging neuron activation with their sensitivity to observational distribution. Formally, our reset process involves injecting controlled perturbation noise into input observations and analyzing output variations across the sub-network layers. Neurons exhibiting significant output changes are identified as highly sensitive and retained for structural exploration. This scoring method effectively reactivates underutilized neurons, addressing the expressivity challenges inherent to sparse policies, and significantly enhancing *plasticity* and capacity to adapt new skills in structure-based continual RL policies.

**Definition 4.1** (Sensitivity dormant scores). Let $\boldsymbol{y}_{k,(i)}^{(l)}(\boldsymbol{s})$ denote the $i$-th neuron output of layer-$l$ given observation $\boldsymbol{s}$ as the input, and $\Delta$ be noise vector to perturb $\boldsymbol{s}$. Given a observation distribution $\mathcal{D}_{\boldsymbol{s}}$ and $\boldsymbol{s} \in \mathcal{D}_{\boldsymbol{s}}$, the sensitive-dormant score of neuron $i$ at layer-$l$ is defined as:

$$c_i^{(l)} = \frac{\mathbb{E}_{\boldsymbol{s} \in \mathcal{D}_{\boldsymbol{s}}} \left| \boldsymbol{y}_{k,(i)}^{(l)}(\boldsymbol{s}) - \boldsymbol{y}_{k,(i)}^{(l)}(\boldsymbol{s} + \Delta) \right|}{\frac{1}{\left\| \phi_k^{(l)} \right\|_1} \sum_j \mathbb{E}_{\boldsymbol{s} \in \mathcal{D}_{\boldsymbol{s}}} \left| \boldsymbol{y}_{k,(j)}^{(l)}(\boldsymbol{s}) - \boldsymbol{y}_{k,(j)}^{(l)}(\boldsymbol{s} + \Delta) \right|}. \tag{8}$$

We say a neuron $i$ in layer $l$ is $\tau$-**dormant** if $c_i^{(l)} \leq \tau$.

**Periodically Resetting.** At the beginning of training, we store the randomly initialized values of all parameters. We periodically evaluate the sensitivity dormant scores for all neurons at fixed training intervals where the scores are computed according to Equation 8. As illustrated in Figure 3, neurons with scores $c_i^{(l)} \leq \tau$ are designated as dormant. Only the **trainable** *task-specific* parameters connected to these dormant neurons are **reset** to their initial stored values. In contrast, all **frozen** parameters are maintained **unchanged**, irrespective of their connection with dormant neurons.

## 5 EXPERIMENTS

### 5.1 EXPERIMENTAL SETTINGS

**Benchmarks.** To assess the performance of SSDE, we follow the standard Continual World experimental setup from (Wolczyk et al., 2022) and conduct extensive evaluations. Our primary benchmark is CW10 from Continual World (Wołczyk et al., 2021), which features 10 representative manipulation tasks drawn from Meta-World (Yu et al., 2019). Additionally, we also use CW20, a version of CW10 repeated twice, to evaluate the transferability of the learned policy across repeated tasks. Details on the CW benchmark is presented in Appendix A.5.

**Evaluation Metrics.** We employ three key metrics, as introduced by Wołczyk et al. (2021): (1) *Average Performance* ($P$) ($\uparrow$): the average performance for all tasks, $P(t) = \frac{1}{|\mathcal{T}_{cl}|} \sum_{k=1}^{|\mathcal{T}_{cl}|} p_k(t)$, where $p_k(t)$ is the success ratio of the $k$-th task at step $t$. (2) *Forgetting* ($F$) ($\downarrow$): the average loss in performance across all tasks after learning is complete, $F = \frac{1}{|\mathcal{T}_{cl}|} \sum_{k=1}^{|\mathcal{T}_{cl}|} [p_k(k \cdot \delta) - p_k(|\mathcal{T}_{cl}| \cdot \delta)]$, where $\delta$ represents the number of environment steps allocated for each task. (3) *Forward Transfer* ($FT$) ($\uparrow$): the transfer is computed as a normalized area between the training curve of the compared method and the training curve of a single-task reference method trained from scratch (no adaptation). The reference performance is denoted as $p_k^b \in [0, 1]$, and the forward transfer is measured as:

$$FT_k := \frac{\text{AUC}_k - \text{AUC}_k^b}{1 - \text{AUC}_k^b}, \quad \text{AUC}_k := \frac{1}{\delta} \int_{(k-1)\cdot\delta}^{k\cdot\delta} p_k(t)\, dt, \quad \text{AUC}_k^b := \frac{1}{\delta} \int_0^{\delta} p_k^b(t)\, dt. \quad (9)$$

**Training Details.** To ensure the reliability and comparability of our experiments, we follow the training details outlined in (Wołczyk et al., 2021), implementing all baseline methods using Soft Actor-Critic (SAC) (Haarnoja et al., 2018). To ensure a fair comparison across tasks, we limit the number of environment interaction steps to 1e6 per task, with each result averaged over five random seeds. And the $Delta$ is defined as 0.01 times the average state over the preceding 1,000 steps. Additional implementation details for SSDE are presented in Appendix A.1.

## 5.2 EVALUATION OF SPARSE PROMPTING-BASED SUB-NETWORK CO-ALLOCATION

We begin with a proof-of-concept experiment to demonstrate the advantage of our proposed network allocation strategy, **sparse prompting with fine-grained co-allocation**, over the sparse prompting in CoTASP. SSDE's co-allocation not only captures task similarity for high-quality $\theta^{fw}$ but also ensures an adequate allocation of task-specific $\theta^{sp}$ to effectively learn new knowledge, significantly enhancing the expressivity of the sparse network.

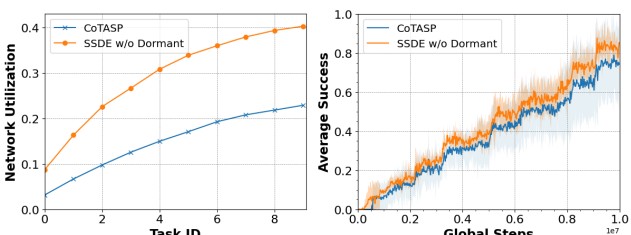

Figure 4: Evaluation on SSDE's co-allocation vs. CoTASP's sparse prompting on CW10-v1. **Left: network utilization**; **Right: average performance**.

Figure 4 illustrates the network utilization ratio under our method and CoTASP, measured as $\frac{\#\text{trained\_parameters}}{\#\text{total\_parameters}}$. Our method achieves a much higher utilization ratio, using nearly $40\%$ of parameters compared to CoTASP's $< 25\%$, reducing parameter waste. Additionally, SSDE consistently outperforms CoTASP in success ratio, highlighting that co-allocation generates sub-networks with greater capacity, leading to improved *plasticity*.

We also examine the computational efficiency of our method compared to its closest structure-based counterparts, PackNet and CoTASP. Table 1 reports the per-task sub-network allocation time. The overhead for PackNet is due to its computationally intensive network pruning (i.e., fine-tuning process with a significant amount of data) and that for CoTASP stems

Table 1: Allocation efficiency.

| Method | Allocation Time ($\downarrow$) |
|---|---|
| CoTASP | 72.2s (6.45$\times$) |
| PackNet | 422.0s (37.68$\times$) |
| **SSDE (Ours)** | **11.2s (1$\times$)** |

from dictionary learning and gradient-based optimization. As a result, PackNet requires more than $37\times$ over SSDE, and CoTASP takes more than $6\times$. These results highlight that SSDE generates high-quality sub-networks with significantly greater computational efficiency.

## 5.3 EVALUATION ON CONTINUAL WORLD BENCHMARK

We conduct benchmark evaluations on the Continual World 10 Tasks (CW-10) & 20 Tasks (CW-20) environments, with results presented in Table 3. Overall, SSDE demonstrates a superior success ratio on CW10-v1, improving the state-of-the-art record of 86%, held by a strong rehearsal-based baseline ClonEx-SAC, to 95%, marking a 9% increase. Figure 6 illustrates the learning curve for SSDE alongside representative baselines. The curve shows a clear advantage for SSDE compare to strong structure-based counterparts like PackNet and CoTASP.

Table 3: Benchmark evaluation results on Continual World (v1).

| Benchmarks-v1 | | CW 10 | | | CW 20 | | |
|---|---|---|---|---|---|---|---|
| Metrics | | $P$ (↑) | $F$ (↓) | $FT$ (↑) | $P$ (↑) | $F$ (↓) | $FT$ (↑) |
| Reg | L2 (Kirkpatrick et al., 2017) | 0.42 ±0.10 | 0.02 ±0.02 | -0.57 ±0.20 | 0.43 ±0.04 | 0.02 ±0.01 | -0.71 ±0.10 |
| | EWC (Kirkpatrick et al., 2016) | 0.66 ±0.05 | 0.03 ±0.02 | 0.05 ±0.07 | 0.60 ±0.03 | 0.03 ±0.03 | -0.17 ±0.07 |
| | MAS (Aljundi et al., 2018) | 0.59 ±0.03 | -0.02 ±0.01 | -0.35 ±0.07 | 0.50 ±0.02 | 0.00 ±0.01 | -0.52 ±0.05 |
| | VCL (Nguyen et al., 2018) | 0.58 ±0.04 | -0.02 ±0.01 | -0.43 ±0.13 | 0.47 ±0.02 | 0.01 ±0.02 | -0.48 ±0.08 |
| | Fine-tuning | 0.12 ±0.00 | 0.72 ±0.02 | 0.32 ±0.03 | 0.05 ±0.00 | 0.72 ±0.03 | 0.20 ±0.03 |
| Struc | PackNet (Mallya & Lazebnik, 2018) | 0.83 ±0.04 | 0.00 ±0.00 | 0.21 ±0.05 | 0.80 ±0.01 | 0.00 ±0.00 | 0.18 ±0.03 |
| | HAT (Serrà et al., 2018) | 0.68 ±0.12 | 0.00 ±0.00 | – | 0.67 ±0.08 | 0.00 ±0.00 | – |
| | CoTASP (Yang et al., 2023)[1] | 0.73 ±0.11 | 0.00 ±0.00 | -0.21 ±0.04 | 0.74 ±0.03 | 0.00 ±0.01 | -0.19 ±0.02 |
| Reh | Reservoir | 0.29 ±0.03 | 0.03 ±0.01 | -1.11 ±0.08 | 0.12 ±0.03 | 0.07 ±0.02 | -1.33 ±0.08 |
| | A-GEM (Chaudhry et al., 2019) | 0.14 ±0.05 | 0.73 ±0.01 | 0.28 ±0.01 | 0.07 ±0.02 | 0.70 ±0.01 | 0.13 ±0.03 |
| | ClonEx-SAC (Wolczyk et al., 2022) | 0.86 ±0.02 | 0.02 ±0.02 | 0.44 ±0.02 | **0.87** ±0.01 | 0.02 ±0.01 | 0.54 ±0.02 |
| MT | MTL (Yu et al., 2019) | 0.51 ±0.10 | – | – | 0.51 ±0.11 | – | – |
| | MTL+PopArt (Hessel et al., 2019) | 0.66 ±0.04 | – | – | 0.65 ±0.03 | – | – |
| | **SSDE (Ours)** | **0.95** ±0.02 | 0.00 ±0.00 | 0.30 ±0.02 | **0.87** ±0.02 | 0.00 ±0.00 | 0.29 ±0.02 |

Additionally, we demonstrate our method could enhance *plasticity* by showing the forward-transfer effect in Figure 7, which compares SSDE and CoTASP to a standard single-task policy provided by Continual World. The results highlight that SSDE achieves stable policy learning progress, converging to higher success ratio with positive forward-transfer (*plasticity*).

We also demonstrate SSDE's scalability in handling more tasks through CW20-v1 experiments. SSDE achieves a comparable performance to ClonEx-SAC, a strong behavior-cloning baseline. It's important to note that CW20 repeats CW10 twice, and ClonEx-SAC would gain access to all expert data and policies for **all CW20 tasks**, resembling offline RL. Our method treats each task as a *new* task, and advances the best score for structure-based method from 80% to 87%. To further illustrate the consistency of SSDE's performance, we evaluated it on CW10-v2. As shown in Table 2, SSDE significantly outperforms its structure-based counterparts.

Table 2: Evaluation on CW10-v2.

| Method | Average Success (↑) |
|---|---|
| CoTASP | 0.73±0.13 |
| PackNet | 0.82±0.04 |
| **SSDE (Ours)** | **0.87±0.03** |

To better assess the quality of the sub-networks generated by SSDE, we provide visualization of the similarity heatmaps of sub-network masks allocated by SSDE in Figure 5. For tasks with similar task embeddings (e.g., Task-2 vs. Task-4 and Task-2 vs. Task-7), we notice strong alignment between the two similarity heatmaps. This demonstrates that SSDE effectively captures task similarities encoded in the descriptions. The strong alignment is crucial for fast adaptation and enhanced *plasticity*, as SSDE can allocate forward-transfer parameters from similar tasks, allowing new tasks to leverage high-quality parameters trained on previous tasks. Additional visualizations of the sub-network mask $\phi^{(l)}$ for each layer are provided in Figure 13 in the Appendix.

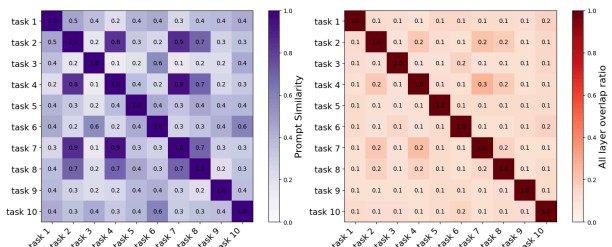

Figure 5: Visualization of **task description similarity** (*left*) and that for **sub-network similarity** (*right*) for SSDE.

## 5.4 ABLATION STUDY

In table 4 presents the results of the ablation study on the CW10-v1 sequence, using average success as the evaluation metric. Among the three SSDE variants: 'w/o $\beta$' does not utilize the Trade-off Coefficient mechanism, keeping all frozen parameters at their original values during training; 'w/o Dormant' removes the reset parameters mechanism; and 'w/o Fine-Grained $\phi, \Psi$' relies solely on a fixed dictionary for sub-network allocation. Additionally, we create an ablation baseline 'W/o Dormant, w/o $\beta$' which employs only co-allocation mask, for a fair comparison with the sparse prompting-based allocation from CoTASP. We

[1]Reproduced from https://github.com/stevenyangyj/CoTASP

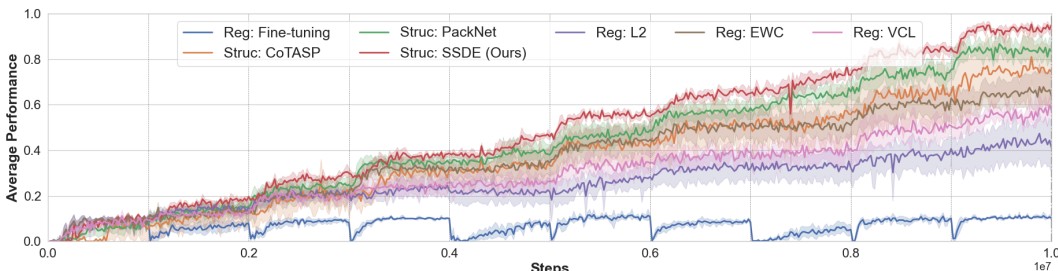

Figure 6: Learning curves for each method on CW10-v1.

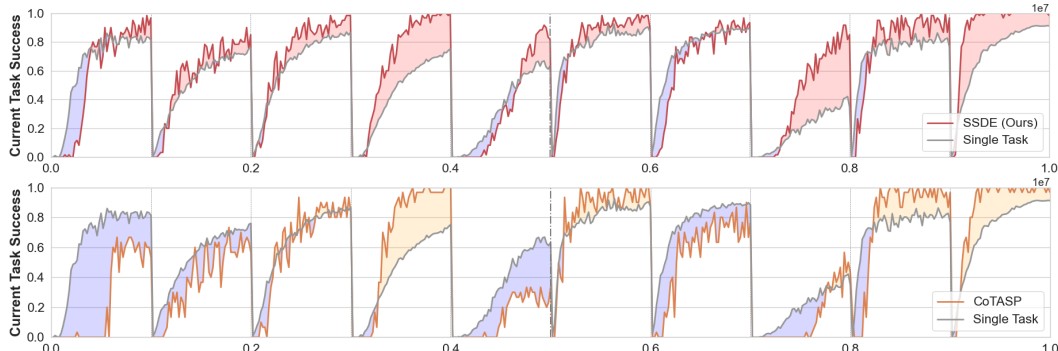

Figure 7: Illustration of **forward transfer** (*plasticity*). Each row compares the learning curve for each CW10-v1 task, against a *single-task* SAC baseline. **Blue** region: single-task learns faster (*negative* transfer); **Red/Yellow** region: *positive* transfer, indicating high plasticity. Overall, SSDE demonstrates strong *plasticity*, consistently leading to faster and more effective learning than SAC.

also introduce a basleine 'SSDE(w/ ReDo)' to compare the effectiveness of our sensitivity-guided dormant score to the original dormant mechanism from ReDo (Sokar et al., 2023).

From this table, we observe that the removal of Fine-Grained mask allocation has the most significant impact on the experimental results. Compared to the sparse prompting in CoTASP, our co-allocation improves the performance by more than 10%. This underscores the critical role of ensuring a dedicated allocation of trainable parameters to incorporate new knowledge, an aspect that has been largely overlooked in previous works. We also observed that the Trade-off Coefficient $\beta$ contributes to more stable ex-

Table 4: Ablation study on CW10-v1.

| Method | Average Success (↑) |
|---|---|
| w/o $\beta$ | $0.83 \pm 0.14$ |
| w/o Dormant | $0.85 \pm 0.06$ |
| w/o Fine-Grained $\phi, \Psi$ | $0.80 \pm 0.07$ |
| w/o Dormant, w/o $\beta$ | $0.81 \pm 0.08$ |
| SSDE (w/ ReDo) | $0.88 \pm 0.02$ |
| **SSDE (ours)** | $\mathbf{0.95} \pm 0.02$ |

perimental results. This is due to its ability to effectively alleviate the impact of frozen parameters on model performance, leading to more consistent outcomes. Additionally, comparing SSDE's sensitivity-guided dormant with ReDo underscores the importance of connecting the sensitivity at observation level to the neuron's activation to address rigidity of sparse policy with particularly constrained trainable capacity. Overall, the core components of SSDE, each addressing critical aspects of continual RL, work synergistically to form the foundation of the model's success.

## 6 CONCLUSION

We introduce SSDE, a novel structure-based continual RL method. At its core, SSDE features an efficient co-allocation algorithm, uniquely allocates dedicated capacity for *trainable* parameters for task-specific learning while leveraging *frozen* parameters for effective forward transfer from previous policies, balanced by a trade-off parameter for fine-grained inference. To address the expressivity limitations of sparse sub-networks, SSDE introduces a structural exploration strategy with sensitivity-guided dormant neurons. These generalizable techniques provide a solid foundation for advancing multi-task, continual learning, and continuous control problems with neural policy. Looking ahead, there is significant potential to further enhance structural sparsity through more dedicated sub-network allocation strategies. Integrating advanced neuron permutation strategies like differentiable wiring mechanisms also offer a promising direction for enhancing the expressiveness of policy.

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

# A   APPENDIX

This supplementary material is organized as follows:

- Sec A.1: implementation details for **SSDE**.
- Sec A.2: a detailed algorithm for **SSDE**.
- Sec A.3: extended discussion on related works.
- Sec A.4: additional experimental results, including (i) a case study on input sensitivity with a complex task $T_5$ : *stick-pull*, (ii) visualization on sub-network similarities, and (iii) per-task score for the experiments.
- Sec A.5: an overview of the robotic manipulation task in Continual World, highlighting the difference between v1 and v2 environments.

The code for reproducing all the experiments and the learning curves will be released after the paper is accepted.

## A.1   IMPLEMENTATION DETAILS

Table 5: Detailed hyperparameter configurations for **SSDE**.

| Hyperparameter | Value | Range |
|---|---|---|
| **SAC** | | |
| Actor hidden size | 1024 | $\{256, 512, 1024\}$ |
| Critic hidden size | 256 | $\{256, 512, 1024\}$ |
| # of hidden layers for meta policy | 4 | $\{2, 3, 4\}$ |
| # of hidden layers for critic $Q_1$ | 4 | $\{2, 3, 4\}$ |
| # of hidden layers for critic $Q_2$ | 4 | $\{2, 3, 4\}$ |
| Activation function | LeakyReLU | - |
| Batch size | 256 | $\{64, 128, 256\}$ |
| Discount factor | 0.99 | - |
| Target entropy | $-2.0$ | - |
| Target interpolation | $5 \times 10^{-3}$ | - |
| Replay buffer size | 1e6 | $\{2e5, 5e5, 1e6\}$ |
| Exploratory steps | 1e4 | - |
| Optimizer | Adam | - |
| Learning rate | $3 \times 10^{-4}$ | - |
| **Continual Learning** | | |
| Training steps for each task | 1e6 | |
| Evaluation steps for each task | | |
| **SSDE** | | |
| Sparsity ratio $\lambda_\Gamma, \lambda_\Lambda$ | $10^{-3}$ | $\{10^{-2}, 10^{-3}, 10^{-4}, 10^{-5}\}$ |
| Trade-off parameter $\beta$ | 0.3 | $\{0.1, 0.2, 0.3, 0.4, 0.5, 0.6, 0.7, 0.8, 1\}$ |
| Dormant threshold $\tau$ | 0.6 | $\{0.2, 0.4, 0.6, 0.8\}$ |
| Dormant reset interval | $8e4$ | $\{1e4, 2e4, 5e4, 8e4, 1e5, 2e5\}$ |

SSDE is developed on top of the Jax Implementation of SAC from JaxRL[2]. The actor policy and critic networks are parameterized as standard MLPs. Notably, SSDE introduces no additional trainable parameters compared to SAC. All calibration masks are determined beforehand through the co-allocation strategy, ensuring that during training, the masks remain fixed, allowing for efficient learning with pre-allocated sub-networks. As a result, SSDE achieves highly computationally efficient sub-network allocation. SSDE also does not employ task-specific policy heads. Additionally, we do not store any data from previous tasks or perform rehearsal on past experiences, differentiating it from rehearsal-based approaches like ClonEx-SAC.

---

[2]https://github.com/ikostrikov/jaxrl

For the evaluation on computational efficiency presented in Table 1, we use a GPU server with $4$ L40, and 120 cores "AMD EPYC 9554P 64-Core Processor" CPU.

## A.2 DETAILED ALGORITHM

---

**Algorithm 1: SSDE**: **S**tructured **S**parsity with **D**ormant Neuron-guided **E**xploration Algorithm

---

**Input:** Meta-policy network $\pi_\theta$, task descriptions $\{S_k\}_{k=1}^N$, trade-off parameter $\beta$, sparse coding $\lambda_{k[\Gamma]}$ and $\lambda_{k[\Lambda]}$, dormant threshold $\tau$, critics $Q_1$ and $Q_2$, temperature $temp$, replay buffer $\mathcal{B} = \emptyset$, training budget $I_\theta$, dormant interval $I_D$, batch_size $bs$.

**Output:** Policy $\theta$, sub-network mask $\{\phi_1, ..., \phi_N\}_{l=1}^L$, P ($\uparrow$), F ($\downarrow$), FT ($\uparrow$).

1 **function** Fine_Grained_CoAllocation
2     **for** *task $k = 1 \rightarrow N$* **do**
3        Compute task embedding $e_k$ = SENTENCE_BERT($S_k$).
4        **for** *layer $l = 1 \rightarrow L$* **do**
5           **if** *k=1* **then**
6              Initialize $\boldsymbol{D}^{(l)} \sim \mathcal{N}(0, 1)$.                 // Fixed dictionary.
7           **end**
8           Solve $\alpha_{k[\Gamma]}^{(l)}$ by sparse_coding $(e_k, \boldsymbol{D}^{(l)}, \lambda_{k[\Gamma]})$ from Eq (3).     // $\Gamma$-Allocation: enhance forward transfer mask
9           Discretize $\phi_{k[\Gamma]}^{(l)} \leftarrow \rho(\alpha_{k[\Gamma]}^{(l)})$
10           Initialize $\boldsymbol{D}_k^{(l)} \sim \mathcal{N}(0, 1)$.                 // Random dictionary.
11           Solve $\alpha_{k[\Lambda]}^{(l)}$ by sparse_coding $(e_k, \boldsymbol{D}_k^{(l)}, \lambda_{k[\Lambda]})$ from Eq (4).     // $\Lambda$-Allocation: enhance trainable capacity
12           $\phi_k^{(l)} = \phi_{k[\Gamma]}^{(l)} \vee \phi_{k[\Lambda]}^{(l)}$.
13        **end**
14     **end**
15     **return** $\{\phi_k^{(l)}\}_{k=1}^N$ *for $l = 1, ..., L$.*
16 **end**
17 **function** Train_and_Eval
    /* Sub-network Allocation Before Training.                           */
18     Get $\{\phi_k^{(l)}\}_{k=1}^N$ for $l = 1, ..., L$ from Fine_Grained_CoAllocation
19     **for** *task $k = 1 \rightarrow N$* **do**
20        Get $\theta_k$ from $\theta$ under neuron mask $\{\phi_k^{(l)}\}_{l=1}^L$.
21        Save current parameters $\theta_{init,k} = \theta$.
22        Computer param mask $\{\tilde{\Psi}_k^{(l)}\}_{l=1}^L$ from neuron mask $\{\phi_k^{(l)}\}_{l=1}^L$ follow Eq (5).
23        **for** $i = 1$ *to $I_\theta$* **do**
24           Act with $a_t \sim \pi_{\theta_k}(s_t; \beta, \Psi_{k-1}, \tilde{\Psi}_k, \phi_k)$ follow Eq (6).     // Forward path
25           Fill $(s_t, a_t, r_t, s_t', \text{done})$ to buffer $\mathcal{B}$.
          /* Learning                                                  */
26           Sample $\tau = \{s_j, a_j, r_j, s_j', \text{done}\}_{j=1}^{bs}$ from $\mathcal{B}$.
27           Computer gradient $g_k$ by optimizing SAC loss.
28           Update $\pi_k$: $\theta_k \leftarrow \theta_k - \alpha(1 - \psi_k) \otimes g_k$, following Eq (7).     // Backward path
29           Update $Q_1, Q_2$ and $temp$ by SGD.
30           **if** $i \% I_D = 0$ **then**
             /* Dormant exploration                                      */
31              Compute $c_i^{(l)}$ for each $\phi_k^{(l)}$ from layer $1, .., L$, following Eq (8).
32              Reset params in $\theta_{sp}$ that is connected to neurons with $c_i^{(l)} \leq \tau$ from $\theta_{init,k}$.
33           **end**
34        **end**
       /* Evaluation                                                        */
35        Evaluate $\pi_{\theta_k}$ on $\mathcal{T}_1, ... \mathcal{T}_N$.
36        $\hat{\Psi}_k^{(l)} = \vee(\Psi_k^{(l)}, \tilde{\Psi}_{k-1}^{(l)})$ for $l = 1, ... L$.     // Accumulate gradient mask
37     **end**
38     **return** $\theta_N$, P ($\uparrow$), F ($\downarrow$), FT ($\uparrow$)
39 **end**

---

### A.3 Detailed Review of Related Works

**Rehearsal-based methods** replay stored experiences from previous tasks to reinforce the stability of neural policies while accommodating adaptation during new task learning. **A-GEM** (Chaudhry et al., 2019) stores samples from previous tasks in a memory buffer and projects gradients from new tasks in a way that prevents interference with past knowledge. **Perfect Memory** (Wołczyk et al., 2021) retains the entire buffer across tasks, ensuring that no information is forgotten by continuously remembering all past data. **ClonEx-SAC** (Wolczyk et al., 2022) disentangles the policy into shared and task-specific components and performs behavior cloning based on previous samples to mitigate forgetting. **Fine-tuning** (Wołczyk et al., 2021) is a forgetful method that continually updates the model with new task data, without specifically managing *stability*. While easy to implement with a relatively straightforward stability-enhancing mechanism, rehearsal-based methods often experience performance degradation as the number of tasks increases, largely due to escalating memory and computational demands.

**Regularization-based methods** attempt to mitigate catastrophic forgetting by introducing constraints or penalties during the learning process, ensuring that important parameters for previously learned tasks are preserved. **EWC** (Kirkpatrick et al., 2016) uses the Fisher information matrix to approximate the importance of each weight and selectively penalizes changes to those deemed crucial for previously learned tasks. **L2** (Kirkpatrick et al., 2017) emposes a $L_2$ penalty to regularize the change of parameters. **MAS** (Aljundi et al., 2018) evaluates a weighted penalty based on the importance of each parameter to the network output. **VCL** minimizes the KL divergence between the prior and posterior of parameters to enhance stability. Despite their effectiveness, a key drawback of these methods is that their rigid regularizers can overly constrain the model, diminishing plasticity while still not fully ensuring the protection of crucial parameters for stability.

**Structure-based methods**, also known as parameter isolation methods, focus on preserving and updating the network architecture to accommodate different tasks. **PackNet** (Mallya & Lazebnik, 2018) performs computational intensive pruning after the training of each task, allocating distinct sub-networks for each task in a sparsity-driven manner. **HAT** (Serrà et al., 2018) learns a hard attention mask for each task to generate sub-networks. Our work is mostly related to the sparse prompting methods driven by task descriptions. **TaDeLL** (Rostami et al., 2020) proposes a coupled dictionary optimization to generate new task parameters as a sparse linear combination over a shared basis with textual descriptions. **CoTASP** (Yang et al., 2023) combines sparse encoding with dictionary learning to generate sparse sub-networks. Though both works pursue sparse sub-networks, their allocation considers allocating the entire sub-network as a monotonic process, while **SSDE** considers fine-grained allocation to accommodate *forward-transfer* parameters and *free*-parameters.

### A.4 Additional Results

#### A.4.1 SSDE's Sensitivity-guided Dormant vs ReDo: A Case Study

Sparse sub-policies in structure-based continual RL methods often encounter the challenge of **rigidity**, where limited trainable parameters progressively lose sensitivity to input variations. This rigidity significantly hinders the policy's ability to adapt to complex tasks requiring precise, multi-step execution. Traditional approaches like **ReDo** (Sokar et al., 2023), which assess neuron responsiveness based solely on activation scales, inadequately capture this issue, as they overlook the critical role of neuron sensitivity to input variations. In this section, we use **Task-5 Stick-Pull** as a case study to statistically measure and showcase the rigidity of sparse sub-policies in terms of input sensitivity. Our observations reveal that policies with restrictive expressivity, constrained by sparse allocation, fail to respond effectively to critical state inputs. This limitation motivates the development of our **sensitivity-guided dormant scores**, which bridge neuron activation with input sensitivity. By addressing the expressivity challenges inherent in sparse sub-policies, SSDE enhances their responsiveness and adaptability, enabling them to handle such demanding tasks more effectively.

**Task Description:** **Task-5 Stick-Pull** (*Grasp a stick and pull a cup with the stick*) is a relatively complex manipulation task from CW10. It involves multiple sub-skills, which need to be executed in sequence: (1) maneuvering the arm to the stick, (2) picking it up, (3) placing the stick into a hole, and (4) finally pulling the cup to a designated target position. Many continual RL policies struggle with this task, often resulting in sub-optimal policies and failing to achieve desired goal-reaching per-

formance. Unless the agent successfully manages all sub-skills, the policy will score a 0% success rate. While training a large network with SAC can easily yield a 100% success rate, structure-based continual learning methods, such as CoTASP, often perform poorly, frequently resulting in failed policy with a 0% success rate on this task.

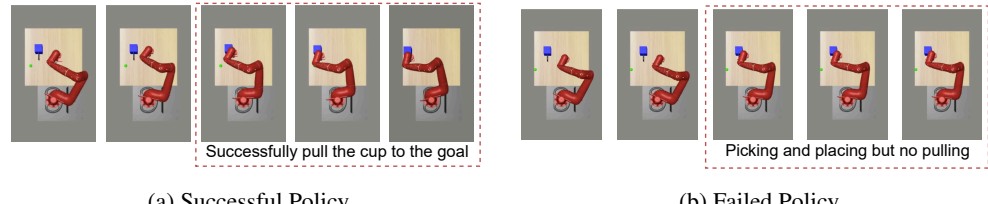

(a) Successful Policy           (b) Failed Policy

Figure 8: Demonstration of the rigidity in the sparse continual RL policy. The failed policy (**right**) stuck in sub-optimal solutions: the agent can *pick up stick*, *place stick to cup*, but cannot *pull cup to goal location*. **The failed policy is highly insensitive to change on goal location from the state inputs.**

**Analysis:** Although the environment provides dense rewards, the failed agent is only able to learn partial skills. As shown in Figure 8, the agent successfully picks up and places the stick but fails to pull it toward the goal. This illustrates a common scenario where a policy becomes stuck in a sub-optimal solution due to insufficient exploration. Enhancing the exploration capabilities of the sparse sub-network remains a critical challenge to address. Additionally, we pose the following assumption:

**Q**: *Could the failure for sparse sub-networks to learn complex manipulation skills be attributed to the **insensitivity of the sub-network parameters to changes in key input features**?*

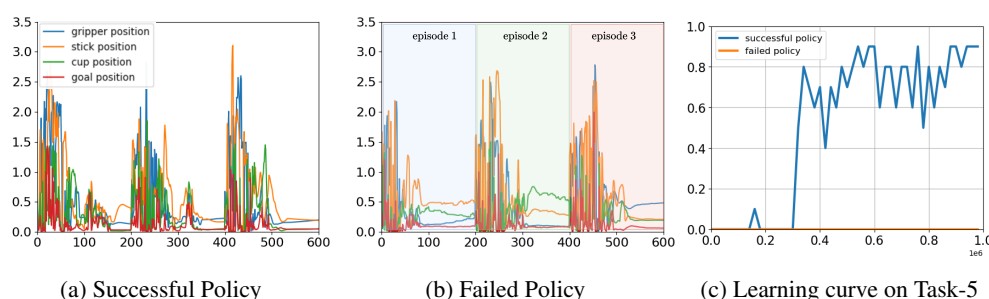

(a) Successful Policy      (b) Failed Policy      (c) Learning curve on Task-5

Figure 9: Evaluation on **input sensitivity** for a successful policy and failed policy, on Task-5. The failed policy remains insensitive to change on **goal position**, compared to other features, and thereby fail in pushing the cup to the goal.

**Empirical Results:** We empirically evaluated the *sensitivity* for sub-network policy parameters w.r.t changes in input. We grouped the input into four major categories (*gripper position, stick position, cup position, and goal position*), and applied $\Delta x$ to each group, where $\Delta x$ is a static perturbation noise to the input. The change in the network's response to the perturbed state is denoted as $a'$, and the difference $|a' - a|$ is recorded. We present the sensitivity of a successful policy and a failed policy, from Figure 9 (a) and (b), respectively. The results show that, while the failed policy exhibits high sensitivity to *cup position* features and *stick position*, its sensitivity to *goal* is significantly lower. In contrast, the successful policy demonstrates more balanced sensitivity across all feature groups, without overlooking certain inputs like goal position. **This motivates us to propose a sensitivity-guided exploration strategy.** We define a novel concept of *sensitivity*-**guided dormant score,** using it to actively identify insensitive parameters in response to input perturbations.

### A.4.2 CO-ALLOCATION FROM SSDE VS. SPARSE PROMPTING FROM COTASP

We discuss on the difference between our proposed sparse prompting method with that from Co-TASP in detail. While CoTASP focuses primarily on *sub-network allocation*, SSDE systematically contributes to: (i) *sub-network allocation*; (ii) *fine-grained inference*; (iii) *training sparse sub-networks with restrictive expressiveness*.

**Allocation: CoTASP** models task similarities with **BERT embeddings**, and applies **sparse coding** to generate sparse allocation parameters ($\alpha$) for each task. These parameters $\alpha$, when discretized, form binary masks applied to neuron outputs, creating sparse sub-networks as policies for each task. However, this allocation strategy has two major drawbacks:

1. **Extensive overlap in $\alpha$ with increasing tasks**: As more tasks are introduced, sub-networks for similar tasks exhibit significantly overlap. This leads to a substantial increase in frozen parameters, thereby reducing the trainable parameters in a policy. This diminishes the policy's capacity to adapt to new tasks.

2. **Iterative updates to $\alpha$ during training**: The sparse prompting masks $\alpha$ are continuously optimized during RL through alternative update. This iterative process incurs significant computational overhead and can lead to training instability.

Given the critical importance of maintaining sufficient capacity for **trainable parameters**, CoTASP addresses this issue by slightly altering the sparse coding's projection plane after each task. This adjustment relies on **dictionary learning**, a computationally intensive and complex optimization process designed to refine the task dictionary.

In contrast, **SSDE**'s co-allocation eliminates the need for computationally intensive **alternative updates** for task prompts ($\alpha$) and **dictionary learning** for task dictionaries. Our method ensures high-quality forward-transfer parameter allocation and sufficient capacity for trainable parameters with a **co-allocation** that combines two processes:

1. **Sparse coding with a *global* shared task dictionary $D$**, which facilitates the allocation of high-quality forward-transfer parameters. This process resembles the sparse coding from CoTASP.

2. **Sparse coding with a *local* task-specific dictionary $D$**, randomly mapping task embeddings to sub-networks.

Combining the two processes effectively resolves the overlapping issue and ensures a **balance between plasticity and stability**. This co-allocation strategy alleviates the need to further optimize task dictionary (remove dictionary learning) or task prompts (remove alternative update), thus operating in a fully **preemptive** manner—computed prior to training with no further updates required. SSDE achieves **higher computational efficiency**.

Beyond **allocation**, SSDE addresses the following crucial limitations in structure-based continual RL methods:

**Inference**: SSDE introduces fine-grained inference, incorporating a novel trade-off parameter to dynamically balance forward-transfer and task-specific parameters.

**Training**: To enhance the expressivity of sparse sub-networks, SSDE proposes a novel sensitivity-guided dormant neuron strategy, improving the adaptability and plasticity of the trained policies.

### A.4.3 SENSITIVITY ANALYSIS

We analyze the sensitivity of key hyperparameters introduced by our method, using experiments conducted with five random seeds. The detailed results are provided in Tables 6 and 7, while Figure 10 illustrates the trends in mean performance. We observe that as the value of $\beta$ increases, the model initially achieves a peak performance at $\beta = 0.3$ with an average success rate of $0.95$, but further increments lead to a decline, accompanied by increased variance, suggesting that excessive reliance on frozen parameters adversely impacts overall performance. On the other hand, for the threshold $\tau$, the model achieves optimal performance at $\tau = 0.6$ with an average success rate of $0.95$. However, higher thresholds such as $\tau = 0.8$ result in a performance drop, likely due to an increased number of reset parameters compromising model stability and reducing the benefits of learned sparsity.

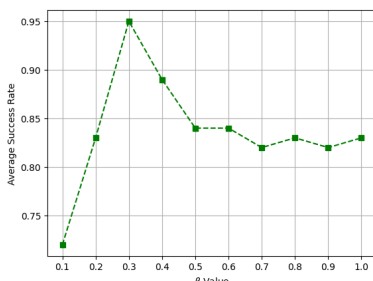 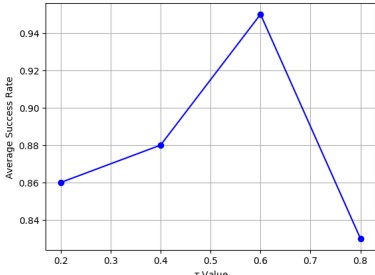

Figure 10: Sensitivity analysis results on crucial model parameters for SSDE. We report the performance of each setting under 5 random seeds. The left figure correspond to the **trade-off coefficient** $beta$, and right one correspond to the **dormant threshold** $\tau$ for sensitivity-guided dormant.

| $\beta$ | 0.1 | 0.2 | 0.3 | 0.4 | 0.5 |
|---|---|---|---|---|---|
| Avg Success | $0.72 \pm 0.01$ | $0.83 \pm 0.02$ | $0.95 \pm 0.02$ | $0.89 \pm 0.03$ | $0.84 \pm 0.05$ |
| $\beta$ | 0.6 | 0.7 | 0.8 | 0.9 | 1.0 |
| Avg Success | $0.84 \pm 0.07$ | $0.82 \pm 0.08$ | $0.83 \pm 0.08$ | $0.82 \pm 0.10$ | $0.83 \pm 0.14$ |

Table 6: Sensitivity analysis results of varying the trade-off parameter $\beta$.

| $\tau$ | 0.2 | 0.4 | 0.6 | 0.8 |
|---|---|---|---|---|
| Avg Success | $0.86 \pm 0.01$ | $0.88 \pm 0.02$ | $0.95 \pm 0.02$ | $0.83 \pm 0.06$ |

Table 7: Sensitivity analysis results of varying the dormant threshold $\tau$.

### A.4.4 GENERALIZATION RESULTS ON BRAX HALFCHEETAH"COMPOSITIONAL" SCENARIO

To demonstrate SSDE can generalize to different problem domains, **we provide extended results on the locomotion tasks, focusing on the ''compositional" task sequence from Brax**. This continual RL task consists of a sequence of four distinct tasks, where the fourth task is a composition of the first and second tasks. This challenging environment not only incorporates compositional skills to complete the task, but also introduces diverse dynamics for the agent.

To integrate SSDE into Brax scenarios, sparse coding works conveniently upon task descriptions as summarized in Table 8. For evaluation, we compare SSDE with two state-of-the-art methods on Brax scenarios, Sun & Mu (2023) and Gaya et al. (2023). The scores for SSDE are averaged across five different random seeds. The overall results in different continual RL metrics are presented in Table 9. We normalize the task returns based on the corresponding SAC-N results, following the approach used in Rewire. The learning curves are shown in Figure 11. **SSDE demonstrates outstanding performance in the Brax compositional scenarios, surpassing the state-of-the-art Performance**

| Task-id | Name | Description |
|---|---|---|
| 1 | tinyfoot | halfcheetah with a tiny foot. |
| 2 | moon | halfcheetah in a small gravity environment. |
| 3 | carry_stuff_hugegravity | halfcheetah is carrying heavy stuff. |
| 4 | tinyfoot_moon | halfcheetah with a tiny foot and in a small gravity environment. |

Table 8: A list of task descriptions for Halfcheetah/compositional scenario from Salina CL (Gaya et al., 2023)

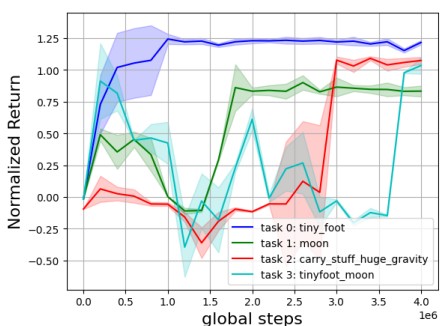

Figure 11: Learning curves for training SSDE on the HalfCheetah-Compositional scenario from Brax, to investigate whether our method could generalize to alternative domains. Each task consists of 1M steps, following the learning sequence: tiny_foot → moon → carry_stuff_huge_gravity → tiny-foot_moon. SSDE achieves an impressive P($\uparrow$) of $1.04 \pm 0.05$, outperforming CSP and Rewire noticeably.

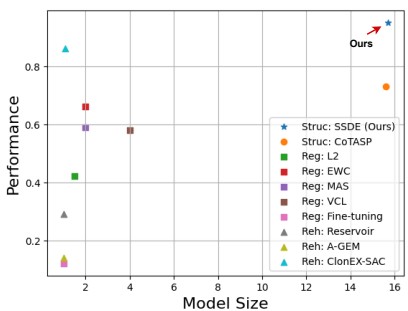

Figure 12: **Trade-off between *model size* (x-axis) and performance (y-axis) on CW10**. Model sizes are normalized relative to the fine-tuning baseline with a hidden size of 256. SSDE and CoTASP share the same network architecture, using a hidden size of 1024. Notably, on modern GPUs, training MLPs with 256 or 1024 hidden units results in negligible differences in both inference and backpropagation times, with a ratio of 1:1.13 for inference and 1:1.07 for backpropagation. In inference, batch size introduces more additional computational overhead, but in JAX's gradient update, batch size has a much smaller impact on backpropagation. Therefore, the gap in backpropagation will be smaller.

($\uparrow$) **achieved by Rewire, with higher Transfer ($\uparrow$) and zero Forgetting ($\downarrow$).** While SSDE operates with a larger model size (hidden dimension of 1024) compared to CSP and Rewire, the parallel computation capabilities of modern GPUs ensure that this increase in size has negligible impact on computation time. This additional results underscore the generalization ability and effectiveness for our proposed method.

Table 9: Comparison of CSP, Rewire, and SSDE in "Halfcheetah/compositional" tasks.

| Method | Performance ($\uparrow$) | Model Size ($\downarrow$) | Transfer ($\uparrow$) | Forgetting ($\downarrow$) |
|---|---|---|---|---|
| CSP | $0.69 \pm 0.09$ | $3.4 \pm 1.5$ | $-0.31 \pm 0.09$ | $0.0 \pm 0.0$ |
| Rewire | $0.88 \pm 0.09$ | $2.1 \pm 0.0$ | $-0.18 \pm 0.09$ | $-0.0 \pm 0.0$ |
| SSDE (Ours) | $1.04 \pm 0.05$ | $15.7 \pm 0.0$ | $0.04 \pm 0.05$ | $0.0 \pm 0.0$ |

From the learning curves, we observe that our method effectively learns tasks without exhibiting catastrophic forgetting. Notably, for the fourth task, we find that during the learning of the first and second tasks, the performance on the fourth task improves to varying degrees during testing. This phenomenon highlights the relevance between tasks, demonstrating that the shared parameters within the model can effectively contribute to learning. These findings provide strong evidence supporting the validity of our shared parameter mechanism.

### A.4.5 MODEL SIZE VS. PERFORMANCE

In this section, we analyze the trade-off between model size and performance across different methods, as shown in Figure 12.

A key feature of our approach is the use of a larger backbone policy network with 1024 neurons, consistent with CoTASP. This larger backbone is essential for structure-based methods like CoTASP and SSDE as it provides the capacity and flexibility needed to allocate parameters effectively and ensure dedicated amount of trainable parameters for each task. In contrast, **many rehearsal-based models utilize a smaller hidden size but incur significant overhead in storing large volumes of**

**data frames**, highlighting a trade-off in resource utilization from a different perspective. Notably, with modern GPUs, **training time doesn't scale linearly with the model size, and reasonable variations in model size (e.g., hidden dimensions of 256 vs. 1024) result in negligible difference in inference or backpropagation update time**, due to the GPU parallelism in computation. To verify this, we measure the inference and backpropagation time for a fine-tuning network architecture (with a relative model size of 1) compared to SSDE's network. Under a standard batch size of 256, the increase in inference/backpropagation time is only about 13%/7%.

### A.4.6 VISUALIZATION OF EACH LAYER'S SUB-NETWORK PROMPTS

We provide a visualization of task similarities encoded in the task embedding $e_t$ and the structural sparse sub-networks allocated by our method, SSDE, for each layer of the policy. The results, shown in Figure 13. Overall, the results reveal several key insights:

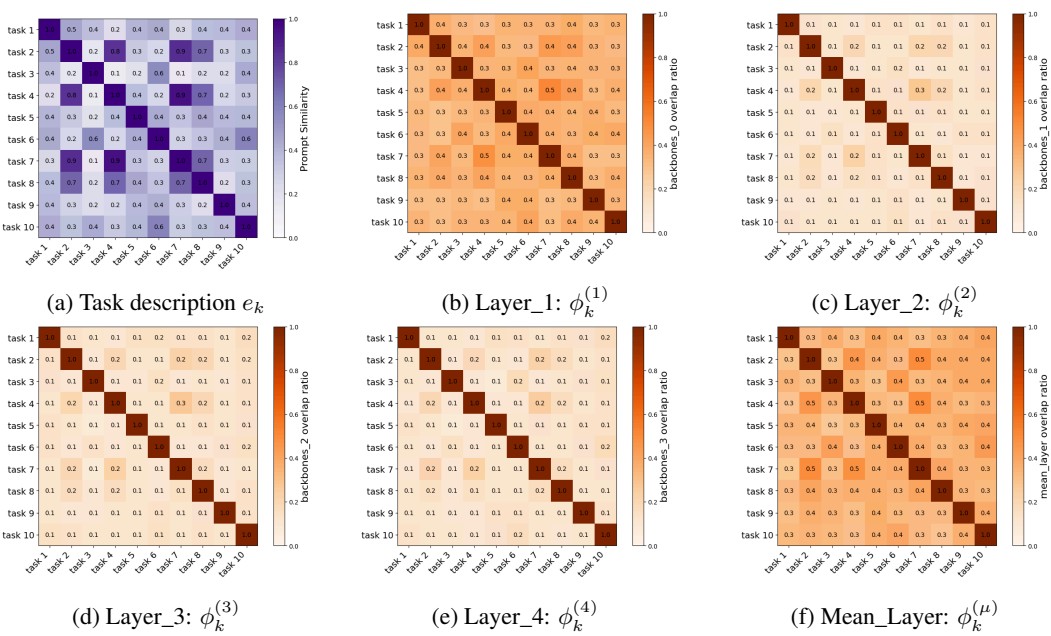

Figure 13: Visualization of task similarities in the task embedding $e_k$ and across each layer from policy allocated by SSDE.

(a) The masks for the input and mean layers are more sensitive to task similarities in the embedding $e_k$, as indicated by the darker colors compared to the hidden layers.

(b) More similar tasks tend to result in more similar sub-network masks. For example, the embedding for Task-2 (*push wall*) shows a strong positive correlation with Task-4 (*push back*) and Task-7 (*push*), i.e., ($e_2$ vs. $e_4$) and ($e_2$ vs. $e_7$) score high from the first sub-figure, and such pattern holds in the sub-network layers compared to other tasks.

(c) The pattern of task similarity remains consistent across different layers of the policy. The darkest blocks, representing the most similar task pairs, appear consistently from Layer_1 through the Mean_Layer.

(d) When allocated sub-network can capture task similarities from $e_t$, it highlights that our method is likely allocating forward-transfer parameters from similar tasks to the current task, facilitating positive forward transfer and enhanced *plasticity*.

| Benchmarks | CW 10 (v1) Success Rate | | | | |
|---|---|---|---|---|---|
| | $T_1$: hammer | $T_2$: push-wall | $T_3$: faucet-close | $T_4$: push-back | $T_5$: stick-pull |
| CoTASP | 0.62 ±0.41 | 0.58 ±0.19 | 0.88 ±0.16 | **1** ±0.00 | 0.32 ±0.41 |
| SSDE (Ours) | **0.98** ±0.04 | **0.92** ±0.08 | **0.94** ±0.09 | **1** ±0.00 | **0.9** ±0.09 |
| | $T_6$: handle-press-side | $T_7$: push | $T_8$: shelf-place | $T_9$: window-close | $T_{10}$: peg-unplug-side |
| CoTASP | **0.96** ±0.05 | 0.66 ±0.11 | 0.32 ±0.25 | **1** ±0.00 | 0.96 ±0.05 |
| SSDE (Ours) | 0.94 ±0.09 | **0.9** ±0.10 | **0.98** ±0.04 | **1** ±0.00 | **0.98** ±0.04 |

| Benchmarks | CW 10 (v2) Success Rate | | | | |
|---|---|---|---|---|---|
| | $T_1$: hammer | $T_2$: push-wall | $T_3$: faucet-close | $T_4$: push-back | $T_5$: stick-pull |
| CoTASP | 0.8 ±0.44 | **0.92** ±0.08 | **0.9** ±0.14 | **0.62** ±0.40 | 0 ±0.00 |
| SSDE (Ours) | **1** ±0.00 | 0.87 ±0.12 | 0.87 ±0.15 | 0.6 ±0.40 | **0.57** ±0.12 |
| | $T_6$: handle-press-side | $T_7$: push | $T_8$: shelf-place | $T_9$: window-close | $T_{10}$: peg-unplug-side |
| CoTASP | **1** ±0.00 | **1** ±0.00 | 0.22 ±0.23 | 0.92 ±0.11 | 0.96 ±0.09 |
| SSDE (Ours) | **1** ±0.00 | 0.87 ±0.06 | **0.87** ±0.06 | **1** ±0.00 | **1** ±0.00 |

Table 10: The success rate (mean±std) scored on each task for *SSDE* and its counterpart *CoTASP* from CW 10 (v1) and CW 10 (v2) benchmarks.

### A.4.7 PER-TASK SCORE FOR THE EXPERIMENTS

### A.5 CONTINUAL WORLD BENCHMARK

Continual World (Wołczyk et al., 2021) is a continual RL benchmark adapted from the Meta-World robotic manipulation tasks (Yu et al., 2019). It features ten distinct manipulation tasks, each varying in aspects such as state space, reward functions, and learning objectives. In the CW10 setup, the ten tasks are presented sequentially to the agent, while CW20 repeats this process twice. The agent is allocated 1 million steps to learn each task. Continual World is designed to evaluate how well RL agents can retain and adapt knowledge as they encounter new tasks in a dynamic, evolving environment.

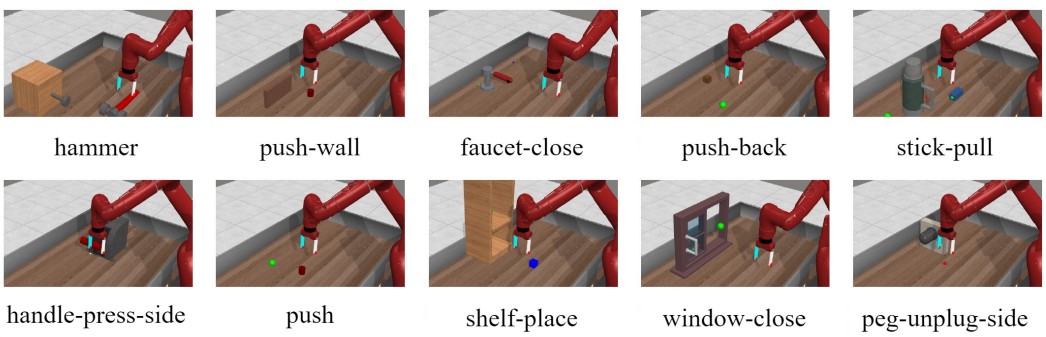

| hammer | push-wall | faucet-close | push-back | stick-pull |
|---|---|---|---|---|
| handle-press-side | push | shelf-place | window-close | peg-unplug-side |

Figure 14: Example states from robotic manipulation tasks in the Continual World benchmark.

**The Continual World benchmark is available in two widely used versions, v1 and v2, with notable differences:**

- **Observation Space**: In v1, the observations were non-Markovian, making it harder for agents to solve tasks. In v2, the observation space has been improved to make it fully Markovian. Additionally, the observation space dimension differs: v1 has a state dimension of 12, while v2 has a state dimension of 39.
- **Reward Structure**: v1 featured a complex and inconsistent reward structure, with rewards differing widely between tasks. In v2, the rewards have been normalized across all tasks, scaled between 0 and 10. This change ensures smoother and more dense rewards, helping agents focus on gradual progress within tasks and making reward interpretation consistent.

Both versions of Continual World are valuable benchmarks for assessing the ability of continual RL agents to handle nonstationary environments. An effective continual policy is expected to consistently excel in its performance on both versions of the benchmark.

| Task-id | Name | Description |
|---------|------|-------------|
| 1 | hammer | Hammer a screw on the wall. |
| 2 | push-wall | Bypass a wall and push a puck to a goal. |
| 3 | faucet-close | Rotate the faucet clockwise. |
| 4 | push-back | Pull a puck to a goal. |
| 5 | stick-pull | Grasp a stick and pull a cup with the stick. |
| 6 | handle-press-side | Press a handle down sideways. |
| 7 | push | Push the puck to a goal. |
| 8 | shelf-place | Pick and place a puck onto a shelf. |
| 9 | window-close | Push and close a window. |
| 10 | peg-unplug-side | Unplug a peg sideways. |

Table 11: A list of task descriptions for Continual World (Yang et al., 2023).

