# OpenReview forum: "Plasticity from Structured Sparsity: Mastering Continual Reinforcement Learning through Fine-grained Network Allocation and Dormant Neuron Exploration"
_ICLR.cc/2025/Conference — Submitted to ICLR 2025_

### Official Review · Reviewer_xXp4 · 2024-10-27

**Soundness:** 3
**Presentation:** 2
**Contribution:** 3
**Rating:** 8
**Confidence:** 3

**Summary:**

This paper proposes a structure-based continual reinforcement learning algorithm. To address the stability-plasticity dilemma in the previous subnetwork allocation approaches, the authors propose a fine-grained allocation strategy with two key designs: (1) The parameter space is decomposed into forward-transfer and task-specific components, which are co-allocated by sparse coding to enhance plasticity. (2) The dormant neurons are periodically reactivated to improve exploration and rapid adaptation in new tasks. The proposed method is validated on the Continual World benchmark and shows significant simprovement in performance.

**Strengths:**

* The paper is generally well written and very informative.
* The technical contributions (fine-grained subnetwork allocation and dormant neuron re-activation) seem solid, both effectively addressing the stability-plasticity dilemma in continual reinforcement learning.
* The experimental results on the Continual World benchmark show great improvements over the existing baselines.

**Weaknesses:**

* The second contribution (dormant neuron re-activation) could be validated more carefully. There are already several works dealing with dormant neurons (e.g. [1, 2]), so a comparison with the existing literature in terms of method differences and experimental results would help to justify the contribution of this paper. Also, since the authors propose a new criterion for dormant neurons, it should be investigated how it overlaps with the original definition in [1], is it possible that it induces false positives and thus destabilizes the learning process?
* The proposed method utilizes task embeddings from a pre-trained BERT, which limits its applicability in broader scenarios involving implicit knowledge that cannot be verbalized. For example, while BERT embeddings work fine in the Continual World benchmark involving manipulation tasks, they may be difficult to transfer to the Brax scenarios used in [3, 4] involving locomotion tasks. It would be interesting to see further experiments or discussions on this issue.
* Regarding efficiency, the authors only discussed their allocation time, but there is no mention of total training time and model size. These are more representative metrics for evaluating the efficiency of continual reinforcement learning, and should be reported in detail in the paper. Also, it would be better if the authors could include an intuitive figure of the performance-size or performance-training time tradeoff, perhaps like Figure 1 in [3] and Figure 4 in [4].
* There is a lack of sensitivity analysis for several hyperparameters, including $\beta$ in Equation (6), $\Delta$ in Equation (8) and $\tau$ for thresholding. The paper determined these hyperparameters by grid search, but I am curious if their choice is critical to the final performance and difficult to choose in practice. The authors could present a sensitivity analysis of these hyperparameters to address my concern, and additional studies of their generalizability across different types of tasks would be appreciated.

---

[1] Sokar, et al. The dormant neuron phenomenon in deep reinforcement learning. ICML 2023.

[2] Dohare, et al. Loss of plasticity in deep continual learning. Nature 2024.

[3] Gaya, et al. Building a subspace of policies for scalable continual learning. ICLR 2023.

[4] Sun, et al. Rewiring neurons in non-stationary environments. NeurIPS 2023.

**Questions:**

* Could the authors provide an intuitive explanation for the use of sparse coding in co-allocation? I agree with the general idea of enhancing plasticity with forward-transfer parameters [1], but I do not fully understand why they can be selected using sparse coding.

---

[1] Lin, et al. TRGP: Trust region gradient projection for continual learning. ICLR 2022.

---

> ### Author Response · Authors · 2024-11-20
> **Response to Reviewer xXp4 (1/4)**
>
> We greatly appreciate Reviewer xXp4's thorough evaluation and constructive suggestions. Please find our response to the reviewer comments below.
>
> > **[Weaknesses] W1(a): How dormant neuron overlaps with original definition in [1]?**
>
> **A:** We wish to clarify that our work is not a straightforward application of dormant neurons proposed in [1] to continual RL domains.  It is important to emphasize that **our sensitivity-guided dormant score is fundamentally distinct from the original dormant score proposed in [1]**. Our new formulation is developed based on a key insight in continual RL: the failure of sparse sub-networks in handling challenging exploration tasks often stems from a loss of sensitivity to the input changes, a critical aspect that is not adequately captured by the original dormant neuron definition.
>
> We illustrate this motivation in detail through an intuitive case study on a hard exploration task, *stick-pull,* from continual-world (Appendix A.4.1). A sub-optimal policy fails to respond to the goal position, a crucial feature for guiding the robot to pull the stick successfully. The original dormant score proposed in [1], only examining the **scale of activation** for neurons, would fail to account for the input sensitivity of policy to input changes. We propose a novel dormant score function to bridges this gap by linking the observation distribution to neuron activation scores, fostering a more effective mechanism for identifying neurons that lack responsiveness to input changes, enabling the sparse sub-network to better adapt to the complexities of continual learning.
>
> Despite of the critical importance of input sensitivity in continual RL, most existing structure-based methods, such as PackNet and CoTASP, focus exclusively on “how to allocate” parameters neglecting “how to enhance the expressiveness of sparse policies” during training. This oversight limits the plasticity of the allocated sub-networks in handling complex and diverse tasks. We believe our work not only significantly extends the capabilities of structure-based continual RL policies but also shifts attention in the field toward a dual focus on “how to allocate” and “how to train with enhanced expressivity.”
>
> **Additional results:** We provide a comparison between our proposed sensitivity-guided dormant score and the original dormant score function from ReDo. Training the sparse sub-network using ReDo’s dormant score formula results in an average success rate that is **7% lower** than that achieved with SSDE in CW10. This highlights the superior effectiveness of our sensitivity-guided dormant score in enhancing the expressiveness of sparse sub-networks for structure-based continual RL.
>
> | Dormant Metric | Average Success |
> | --- | --- |
> | Redo | 0.88$\pm$0.02 |
> | Sensitivity | **0.95$\pm$0.02** |
>
> The discussion is also detailed in the `Appendix A.4.1: SSDE's Sensitivity-Guided Dormant vs. ReDo: A Case Study` in our paper.
>
> > **[Weaknesses] W1(b): Could the new criterion induce false positives and destabilize learning?**
>
> **A:** Different dormant score formulations would identify varying subset of neurons as dormant for resetting. Consequently, there are no definitive ground-truth labels to precisely classify neurons as dormant or to confirm the presence of false positives.
>
> Dormant scores are typically evaluated using finite samples collected from the environment interactions, and we do acknowledge the potential for resetting some active neurons (i.e., false positives). However, based on our empirical investigation (e.g., learning curves in Figure 7), we observe that **the dormant algorithm operates stably without apparent destabilization issues** when the threshold is set appropriately. Using our recommended threshold of $\tau=0.6$, the algorithm **resets approximately only 30% of neurons**, constituting only a small portion of parameters. Consequently, the policy can recover promptly after the reset, as evidenced by the learning curves shown in Figure 7.
>
> Therefore, we conclude that the dormant-based method effectively enhances the expressiveness of sparse sub-networks without compromising the stability of training.
>
> [1] Sokar, Ghada, et al. "The dormant neuron phenomenon in deep reinforcement learning." ICML 2023.

---

> ### Author Response · Authors · 2024-11-20
> **Response to Reviewer xXp4 (2/4)**
>
> > **[Weaknesses] Q2(a): Discussion and comparison with locomotion tasks from the Brax scenarios [1, 2]?**
>
> **A:** We sincerely thank the reviewer for their interest in extending the discussion on BERT embeddings and for highlighting the Brax scenarios and related works [1,2]. Below, we address these points in detail and provide further comparisons and discussions.
>
> **BERT embeddings**:  Each task in Continual World (CW) involves executing a sequence of modular skills that can often be interpreted and expressed in natural language. BERT embeddings are a natural choice for encoding these task descriptions and have proven effective in our sparse coding-based co-allocation strategy. Moreover, our method is flexible and could accommodate other types of embeddings if tasks are better represented through alternative modalities. However, we acknowledge that our approach assumes continual RL tasks share modular **skills components** that can be clearly described in text, a limitation we have now explicitly discussed in the revised manuscript.
>
> **Brax Scenarios**: The key difference between locomotion tasks in Brax and CW lies in task variability. Brax tasks primarily **differ in their dynamics functions**, while CW assumes a fixed robot dynamics model across tasks. Changes in dynamics functions are typically represented numerically, making them difficult to encode meaningfully using textual embeddings. Consequently, our BERT-based approach, designed for tasks with modular and interpretable descriptions, may not be optimal for such problem domains. But SSDE can offer a reasonable solution to it with good capability for mitigating forgetting and operating with high computational efficiency. We have added this discussion to highlight the contextual scope of our method.
>
> **Discussion on [1]**: The Continual Subspace of Policies [1] is a **growing-size approach**, while SSDE is a fixed-size method. Each policy from CSP trains full-size SAC policies for each task, and shares parameters across tasks, treating all parameters equally important in each growing subspace. In contrast, SSDE decomposes a parameter space into sub-policies with task-specific and shared components, ensuring fine-grained control over forward transfer. Additionally, while CSP requires multiple training cycles to compare learning performance across expanded and original subspaces, SSDE trains each sparse sub-policy once, ensuring computational efficiency. Despite the different paradigms, we acknowledge the complementary insights CSP provides and have cited it in the revised paper.
>
> **Discussion on [2]**: Rewire [2] introduces a novel mechanism that permutes the outputs of hidden layers via a differentiable sorting algorithm, **learning multiple wirings to diversify the policy.** Unlike SSDE, Rewire operates on full-size networks and ensures stability by freezing wiring while weights remain adaptable with L2 regularization. While Rewire underperforms SSDE by 11% on CW10, its approach is intuitive and inspiring. We believe that its differentiable wiring concept could complement sparse network-based strategies like ours. Furthermore, integrating Rewire's approach with our sensitivity-guided dormant scores could offer a promising direction for enhancing network expressiveness and stability in future research.
>
> **Revisions**: We have incorporated references to [1] and [2] and expanded discussions on the assumptions and limitations of BERT embeddings, as well as the suitability of SSDE for specific problem domains like Brax scenarios. The overall comparison is shown below and the detailed information can be found in Appendix A.4.4. More updates are highlighted in the revised manuscript.
>
> |  | Performance | Model Size | Transfer | Forgetting |
> | --- | --- | --- | --- | --- |
> | CSP[1] | 0.69$\pm$0.09 | 3.4$\pm$1.5 | -0.31$\pm$0.09 | 0.0$\pm$0.0 |
> | Rewire[2] | 0.88$\pm$0.09 | 2.1$\pm$0.0 | -0.18$\pm$0.09 | -0.0$\pm$0.0 |
> | SSDE | **1.04$\pm$0.05** | 15.7$\pm$0.0 | **0.04$\pm$0.05** | 0.0$\pm$0.0 |
>
> [1] Gaya, Jean-Baptiste, et al. "Building a subspace of policies for scalable continual learning." ICLR 2023.
>
> [2] Sun, et al. "Rewiring neurons in non-stationary environments." NeurIPS 2023.

---

> ### Author Response · Authors · 2024-11-20
> **Response to Reviewer xXp4 (3/4)**
>
> > **[Weaknesses] W2(b): What are the implications of using BERT-based task embeddings on generalizability?**
>
> **A: Advantages**: BERT-based embeddings provide a robust, training-free mechanism to extract semantically rich representations from task descriptions, making them **well-suited for the modular and interpretable skill components** in Continual World. This allows our sparse coding-based co-allocation strategy to identify task relationships and allocate forward-transfer parameters effectively, enabling generalization across tasks with diverse textual descriptions. Additionally, our co-allocation mechanism remains flexible and can work with other types of embeddings, should they better capture task relationships in non-textual domains.
>
> **Limitations**: The approach assumes that tasks in continual RL share modular skills that can be clearly described textually. Poorly written or ambiguous descriptions may hinder the effectiveness of BERT embeddings. As it generally assumes tasks similarities, it is unclear how it deals with conflicting tasks, though most continual RL assumes tasks are related. Furthermore, tasks better represented by non-textual modalities, such as visual or sensor data, may require multi-modal or alternative embedding strategies to maintain generalizability. Future extensions could address this by incorporating domain-specific embeddings.
>
> > **[Weaknesses] W3: Could you provide a figure of the performance-size or performance-training time tradeoff?**
>
> **A:**  Yes, we have added a figure in Appendix 4.5 illustrating the trade-off between model size and performance.
>
> Our method performs well with the same network size as our closest counterpart, CoTASP. The continual RL policy is implemented as an MLP with four fully connected layers, each having a hidden size of 1024, detailed in the hyperparameter configuration in Appendix A.1.
>
> Comparing training times for continual RL methods developed for CW is challenging, as methods with and without experience replay can result in significantly different training durations. Additionally, there has been no prior work on CW that addresses this issue.
>
> To address the reviewer’s concern, **we provide comparison on total training time for SSDE, CoTASP and PackNet** below. A comparison with ClonEX-SAC is not available, as the method is not open-source and the important setting of behavior cloning frequency and replay buffer size for expert data cloning is not explicitly stated in the paper. The result demonstrates our method is computationally efficient, saving 10%+ total training time compared to the other two structure-based counterparts.
>
> | Method | Total Time (hours) |
> | --- | --- |
> | SSDE | 10.05 |
> | CoTASP | 11.08(+10.2%) |
> | PackNet | 11.13(+10.7%) |
>
> Additionally, it is important to highlight that with modern GPUs, **training time does not scale linearly with model size** due to the efficiency of parallel computations. For instance, increasing the hidden layer size from 256 to 1024 results in minimal changes in computational time, with a ratio about 1:1.13 for inference and 1:1.07 for backpropagation in our device. This demonstrates that SSDE's larger model size introduces negligible computational overhead while delivering significant performance improvements.
>
> > **[Weaknesses] W4: Sensitivity analysis for key parameters $\beta$, $\tau$.**
>
> **A:** We appreciate the reviewer’s interest in the sensitivity analysis of key parameters **$\beta$** and **$\tau$**. Below, we provide detailed results demonstrating how variations in these parameters affect performance.
>
> | **Threshold** | 0.2 | 0.4 | 0.6 | 0.8 |
> | --- | --- | --- | --- | --- |
> | average success | 0.86$\pm$0.01 | 0.88$\pm$0.02 | **0.95$\pm$0.02** | 0.83$\pm$0.06 |
>
> | **Trade-off** | 0.1 | 0.2 | 0.3 | 0.4 | 0.5 | 0.6 | 0.7 | 0.8 | 0.9 | 1.0 |
> | --- | --- | --- | --- | --- | --- | --- | --- | --- | --- | --- |
> | average success | 0.72$\pm$0.01 | 0.83$\pm$0.02 | **0.95$\pm$0.02** | 0.89$\pm$0.03 | 0.84$\pm$0.05 | 0.84$\pm$0.07 | 0.82$\pm$0.08 | 0.83$\pm$0.08 | 0.82$\pm$0.10 | 0.83$\pm$0.14 |

---

> ### Author Response · Authors · 2024-11-20
> **Response to Reviewer xXp4 (4/4)**
>
> > **[Question] Q1: Motivation for sparse coding for prompting.**
>
> **A:** Sparse coding's exceptional ability to efficiently **derive compact and interpretable representations** makes it well-suited for addressing the core challenge of structure-based continual RL: **deriving** **sparse sub-networks** that accommodate each task within a shared parameter space while maintaining scalability and computational efficiency.
>
> In the context of network allocation, the problem can be framed as calibrating the output for each layer of a shared neural network policy using neuron-level binary masks. There are two primary strategies for deriving these masks:
>
> 1. **Post-hoc allocation strategies** (e.g., PackNet), which explore sparse sub-networks after policy training through fine-tuning processes such as pruning. While effective, these approaches are computationally expensive due to the additional fine-tuning required.
> 2. **Preemptive allocation strategies** (e.g., CoTASP), which derive sub-network structures prior to policy training, enabling the direct training of sub-policies on predefined sparse structures without the need for subsequent fine-tuning.
>
> Sparse coding is a natural candidate for preemptive allocation strategies, as it enables the derivation of task relationships by leveraging **cross-modal information**, such as textual task descriptions encoded by pre-trained language models like BERT. Our empirical results have showcased this capability, demonstrating that the pairwise task similarities are effectively reflected in the sub-network structures. Specifically, sub-network structure for each layer (Figure 10) and the overall sub-network policy structure (Figure 5) reveal that tasks with similar descriptions are allocated closely related sub-network structures through sparse coding.
>
> The sparse coding-based allocation strategy proposed in our work is fundamentally different from that in CoTASP. While CoTASP performs alternative updates between the sub-network prompting parameter $\alpha$ and policy parameters during RL training, and dictionary learning to optimize the task dictionary, our work offers a fresh perspective. We showcase that high-quality allocation masks can be generated in a **completely preemptive manner.** Additionally, our work introduces a novel intuition of jointly attending to the allocation of forward-transfer parameters while ensuring dedicated capacity for trainable task-specific parameters simultaneously.
>
> This improvement makes our method not only more computationally efficient than the sparse coding-based allocation in CoTASP but also capable of generating sub-networks of better quality (as shown in Figure 4). To the best of our knowledge, our work is the first to propose a completely preemptive sparse coding approach for structure-based continual RL. This novel contribution provides a solid foundation for future advancements in preemptive allocation strategies.

---

> > ### Comment · Reviewer_xXp4 · 2024-11-20
> >
> > Thank you for the additional experiments that addressed all of my concerns. Therefore, I am raising the score from borderline accept to accept.

---

> > > ### Author Response · Authors · 2024-11-21
> > >
> > > Thank you for your thoughtful feedback and for acknowledging our responses. We sincerely appreciate the time and effort you devoted to reviewing our rebuttal. We are delighted that our responses addressed your concerns and met your expectations. We also greatly appreciate your decision to increase the scores.

---

### Official Review · Reviewer_q2Ui · 2024-11-03

**Soundness:** 3
**Presentation:** 3
**Contribution:** 2
**Rating:** 5
**Confidence:** 3

**Summary:**

This work introduces SSDE, a novel structure-based continual RL method. SSDE formulates an efficient co-allocation algorithm that enhances sub-network allocation by increasing the capacity for trainable parameters while leveraging frozen parameters for effective forward transfer from previous policies. SSDE introduces a trade-off parameter to balance these two groups of parameters in its fine-grained inference process.

**Strengths:**

(1)SSDE not only achieves SOTA stability standards but also achieves competitive plasticity even when compared to strong behavior cloning baselines that benefit from data replay.
(2)Experimental results demonstrate that SSDE outperforms current state-ofthe-art methods and achieves a superior success rate of 95% on the CW10-v1.

**Weaknesses:**

As shown in Table 3, SSDE takes no obvious advantages in F and FT metrics. These two metrics usually represent backward and forward transfer. So why Average Performance (P) is so good while F and FT not? I am a little confused. Also, ClonEx-SAC seems to perform comparably with SSDE on CW 20, although it replays data.

**Questions:**

(1)Can we have more continual RL tasks? In the existing version, CW tasks may be not very convincing, especially CW20. Maybe you can refer to [a] for more RL tasks to evaluate continual RL methods.
[a] Online Continual Learning for Interactive Instruction Following Agents, 2024
(2)\beta in Eq.6 is very important since it balance the stability and plasticity in CL. Could you show its sensitivity or how do you decide the optimal value.

---

> ### Author Response · Authors · 2024-11-20
> **Response to Reviewer q2Ui (1/2)**
>
> We sincerely thank Reviewer q2Ui for the detailed and insightful comments. Please find our response (marked as A) to the reviewer comments below.
>
> > **[Weaknesses] W1: Why is $P(\uparrow)$so good while $F(\downarrow)$ and $FT(\uparrow)$ not?**
>
> **A:** $P(\uparrow)$ is widely recognized as the primary metric for continual RL because it directly reflects task performance, which is the key objective to optimize for these algorithms. On the other hand, $F(\downarrow)$ and $FT(\uparrow)$ are supplementary metrics that often reflect trade-offs inherent to different types of continual RL methods.
>
> **$FT(\uparrow)$, in particular, is inherently influenced by access to prior data, which creates a bias favoring rehearsal-based methods.** Rehearsal-based approaches like ClonEx-SAC store and rehearse data from previous tasks, enabling faster learning and higher **$FT(\uparrow)$**, as the metric evaluates the area under the learning curve relative to a standard SAC baseline.
> This bias is especially evident in the CW20 benchmark, where the repeating nature of the task sequence provides ClonEx access to expert data and policies for all tasks after completing the first 10 tasks. This setup allows ClonEx to rehearse and fine-tune its policies with guidance from optimal teachers, resulting in a significant improvement in **$FT(\uparrow)$** compared to CW10.
>
> In contrast, **SSDE treats each incoming task as a new task**, **allocating a separate sub-network through its co-allocation strategy**. Unlike rehearsal-based methods, structure-based methods like SSDE, CoTASP, and PackNet do not store or rehearse prior data, which naturally results in lower **$FT(\uparrow)$**. However, this trade-off is offset by superior performance in **$F(\downarrow)$**, where SSDE achieves a perfect score of **0** on both CW10 and CW20, outperforming ClonEx-SAC in these scenarios.
>
> The differences in methodology and access to prior information highlight why **no single continual RL algorithm simultaneously excels across all metrics**. Each metric reflects distinct aspects of performance, and the algorithms’ designs cater to specific trade-offs between stability, plasticity, and computational efficiency.
> Nevertheless, SSDE demonstrates state-of-the-art performance across **$P(\uparrow)$, $F(\downarrow)$,** and **$FT(\uparrow)$** when compared to its structure-based counterparts. This is achieved through SSDE’s innovative design, which balances stability and plasticity via structural sparsity, co-allocation masks, and sensitivity-guided dormant neurons. These features not only enhance SSDE’s ability to handle diverse tasks but also ensure scalability and computational efficiency, making it a practical and robust solution for continual RL.
>
> > **[Weaknesses] W2: ClonEx perform comparably with SSDE on CW20.**
>
> **A:** The benchmark task CW20 is constructed by **repeating the ten tasks from CW10 twice**.
>
> ClonEx stores both expert data and expert policies from previous tasks. After completing the first 10 tasks, **it gains access to both data and policies for *all tasks from CW20*.** This setup provides ClonEx a significant advantage for handling the repeated tasks under conditions resembling offline RL.
>
> In contrast, SSDE treats the repeated tasks as entirely new tasks to allocate them within its shared parameter space without relying on any stored data or policies from previous tasks.
>
> The performance score on CW20 showcases SSDE’s strong capability to scale effectively with an increased number of heterogeneous tasks. Despite not accessing any previous information, SSDE achieves comparable $P(\uparrow)$ performance to ClonEx, the state-of-the-art method, on CW20. Furthermore, SSDE is significantly more computationally and memory-efficient than ClonEx. This makes SSDE particularly suited for real-world scenarios where memory and computational efficiency are critical, such as robotics or autonomous systems.

---

> ### Author Response · Authors · 2024-11-20
> **Response to Reviewer q2Ui (2/2)**
>
> > **[Questions] Q1: Can we have more continual RL tasks?**
>
> **A:** Yes, we agree that evaluating on additional benchmarks is valuable. While our experiments focused on CW10 and CW20, we also extend our evaluation to benchmarks such as Barx to further validate SSDE's scalability and performance.
>
> We integrate SSDE to the Halfcheetah-compositional continual RL task in Brax. From the results shown below, the performance of SSDE is better than the state-of-the-art baselines CSP and Rewire. Our results highlight the **effectiveness and generality** of SSDE when evaluated across problems with diverse nature. Although **SSDE** requires a larger model, the **performance** and **transfer** improvements make it a compelling choice for continual RL tasks. And the detailed information and learning curve can be found in Appendix A.4.4
>
> |  | Performance | Model Size | Transfer | Forgetting |
> | --- | --- | --- | --- | --- |
> | CSP [1] | 0.69$\pm$0.09 | 3.4$\pm$1.5 | -0.31$\pm$0.09 | 0.0$\pm$0.0 |
> | Rewire [2] | 0.88$\pm$0.09 | 2.1$\pm$0.0 | -0.18$\pm$0.09 | -0.0$\pm$0.0 |
> | SSDE | **1.04$\pm$0.05** | 15.7$\pm$0.0 | **0.04$\pm$0.05** | 0.0$\pm$0.0 |
>
> > **[Questions] Q2: Sensitivity of $\beta$.**
>
> **A:** We provide sensitivity analysis for $\beta$ . Our method performs best with $\beta=0.3$, which we recommend for the main experiments. Notably, $\beta=1.0$ resembles the scenario without a fine-grained tradeoff, resulting in reduced performance. The results demonstrate the crucialness of the trade-off parameter.
>
> | **$\beta$** | 0.1 | 0.2 | 0.3* | 0.4 | 0.5 | 0.6 | 0.7 | 0.8 | 0.9 | 1.0 |
> | --- | --- | --- | --- | --- | --- | --- | --- | --- | --- | --- |
> | Average success | 0.72$\pm$0.01 | 0.83$\pm$0.02 | **0.95$\pm$0.02** | 0.89$\pm$0.03 | 0.84$\pm$0.05 | 0.84$\pm$0.07 | 0.82$\pm$0.08 | 0.83$\pm$0.08 | 0.82$\pm$0.10 | 0.83$\pm$0.14 |
>
> Please refer to `Appendix A.4.3: Sensitivity Analysis` for more detailed sensitivity analysis figures and extended discussions.
>
> [1] Gaya, Jean-Baptiste, et al. "Building a subspace of policies for scalable continual learning." ICLR 2023.
>
> [2] Sun, et al. "Rewiring neurons in non-stationary environments." NeurIPS 2023.

---

> > ### Comment · Reviewer_q2Ui · 2024-11-27
> > **Thank you for your detailed response**
> >
> > Thank you for your detailed response, which I have read carefully. SSDE outperforms other competitors, which raises the question: does this performance benefit from the use of a large model size? Additionally, could you provide more detailed information about the experiments conducted on the Barx dataset?

---

> ### Author Response · Authors · 2024-11-22
> **Gentle Reminder to Review Our Rebuttal**
>
> Dear Reviewer q2Ui,
>
> We are deeply grateful for your thoughtful and constructive feedback on our manuscript. Your comments, particularly regarding the results comparison with ClonEx-SAC, additional experiments, and the sensitivity analysis of the $\beta$ value, are helpful in guiding us to improve our work.
>
> We have been working carefully to address your concerns, and we hope having a further discussion with you that would greatly help us improve the quality and clarity of our work.
>
> We understand how busy you must be, but we kindly wish to remind you of the upcoming discussion period deadline on November 26, 2024 (AoE). We deeply value your expertise and insights, which would greatly help us refine and enhance the quality of our manuscript.
>
> Thank you again for your time and support. It means a great deal to us.
>
> With sincere gratitude,
>
> The Authors

---

> ### Author Response · Authors · 2024-11-24
> **Follow-Up on Rebuttal Feedback**
>
> Dear Reviewer q2Ui,
>
> I hope this message finds you well.
>
> As the discussion deadline of **Nov 26 AoE** approaches, approximately **68 hours** remain. We noticed that we have not yet received your feedback on our rebuttal. Could you please take a moment to review our rebuttal and share your feedback within this time frame? Your insights are crucial for the final evaluation of our submission. We greatly appreciate your expertise and look forward to your valuable comments.
>
> Thank you very much for your attention to this matter.
>
> Best regards,
>
> The Authors

---

> ### Author Response · Authors · 2024-11-28
> **Response to Reviewer q2Ui (1/2)**
>
> We sincerely thank the reviewer for carefully reading our response and providing insightful follow-up comments. We greatly appreciate engaging with your valuable questions on model size.
>
> > SSDE outperforms other competitors, which raises the question: does this performance benefit from the use of a large model size?
>
> To clarify, while the total model size of SSDE is 15.7x due to its architecture, the majority of these parameters remain unused due to our sparse prompting mechanism. **The final model size of SSDE can be reduced if we prune the unused parameters**. Therefore, the fairer comparison should be **trained model size** which is defined as the number of parameters trained at the end of training. The trained model size of SSDE is **4.0x** and only **17.6% larger than CSP**, as shown in the table below:
>
> | Method | Performance (compare to SAC-N) | Total Model Size (# total params / # SAC total params)  | Trained Model Size (# trained params / # SAC total params) |
> | --- | --- | --- | --- |
> | CSP [1] | 0.69$\pm$0.09 | 3.4$\pm$1.5 | 3.4$\pm$1.5 (0.85x) |
> | Rewire [2] | 0.88$\pm$0.09 | 2.1$\pm$0.0 | 2.1$\pm$0.0 (0.53x) |
> | SSDE (original, 1024 hidden size) | 1.04$\pm$0.05 | 15.7$\pm$0.0 | 4.0$\pm$0.02 (1x) |
>
> Table: Comparison of performance and model size on HalfCheetah-Compositional. SAC-N refers to N standalone SAC networks trained on N tasks separately.
>
> Since SSDE uses a **fixed-structure network**, initializing the model with an appropriate hidden size is crucial to preserve sub-network capacity for continual learning. Sparse prompting compresses the parameter space into sparse subspaces, and **this compression requires sufficient pre-allocated capacity to ensure both plasticity and stability during learning**. This challenge of parameter allocation further showcases what SSDE’s co-allocation mechanism is designed to address, enabling the model to achieve superior performance with efficient utilization of trainable parameters.
>
> **Additional results with small model sizes**:
>
> To further address the reviewer’s concern, we conducted additional experiments with reduced hidden sizes on the Brax experiments. The results demonstrate that even with a much smaller hidden size 256, SSDE outperforms CSP by **a noticeable margin of 0.11** while utilizing a **trainable model size approximately 1/4 of CSP**. Additionally, SSDE with hidden sizes of 256 or 512 achieves comparable performance to the state-of-the-art method Rewire on Brax. SSDE with 1024 hidden size significantly outperforms both CSP and Rewire. These findings highlight the efficiency and robustness of SSDE even under constrained model size.
>
> | Method | Performance (compare to SAC-N) | Total Model Size (# total params / # SAC total params)  | Trained Model Size (# trained params / # SAC total params) |
> | --- | --- | --- | --- |
> | CSP [1] | 0.69$\pm$0.09 | 3.4$\pm$1.5 | 3.4$\pm$1.5 |
> | Rewire [2] | 0.88$\pm$0.09 | 2.1$\pm$0.0 | 2.1$\pm$0.0 |
> | SSDE with 1024 hidden size | 1.04$\pm$0.05 | 15.7$\pm$0.0 | 4.0$\pm$0.02 |
> | SSDE with 512 hidden size | 0.83$\pm$0.04 | 4.0$\pm$0.0 | 2.8$\pm$0.03 |
> | SSDE with 256 hidden size | 0.80$\pm$0.04 | 1.1$\pm$0.0 | 0.89$\pm$0.03 |
>
> **Model size vs computational efficiency**:
>
> It is worth emphasizing an important insight about SSDE’s model size and its relation to computational efficiency. While SSDE employs larger total model sizes, its computational requirements remain manageable. Increasing the hidden size from 256 to 1024 results in only a slight increase in computational time, approximately 1:1.13 for inference and 1:1.07 for backpropagation, based on our hardware observations.
>
> In contrast, **growing-size approaches** like CSP dynamically expand the network during training, which can lead to fundamentally different nature in computational efficiency. This additional insight highlights that the increased total model size introduced by sparse prompting minimally impact on the computational cost of training SSDE on continual RL tasks, making it both efficient and scalable.
>
> [1] Gaya, Jean-Baptiste, et al. "Building a subspace of policies for scalable continual learning." ICLR 2023.
>
> [2] Sun, et al. "Rewiring neurons in non-stationary environments." NeurIPS 2023.

---

> ### Author Response · Authors · 2024-11-28
> **Response to Reviewer q2Ui (2/2)**
>
> > Could you provide more detailed information about the experiments conducted on the *Brax* dataset?
>
> We provide comprehensive details about the experimental setup about Brax to ensure transparency. We adopted identical task settings and hyperparameter configurations for the HalfCheetah-Compositional environment as reported in Rewire (Table 2 from appendix in [2]) and CSP (Table 5 from appendix in [1]).
>
> **Task Settings:**
> Our experiments are conducted within the Salina_CL task framework from [1], which leverages the Brax physics engine. Specifically, we utilize the HalfCheetah environment (observation dimension: 18, action dimension: 6). Salina_CL introduces variations in environment parameters to generate a diverse set of tasks, categorized into four scenarios: Forgetting, Transfer, Robustness, and Compositional. For evaluation, we focus on the HalfCheetah-Compositional scenario, a continual learning task sequence with the following progression: tinyfoot → moon → carry_stuff_hugegravity → tinyfoot_moon. Each task is trained over 1 million environment steps.
>
> The task-specific dynamics configurations are detailed in the table below:
>
> | **HalfCheetah** |  |
> | --- | --- |
> | normal | - |
> | tinyfoot | {”foot”: 0.5} |
> | moon | {”gravity”: 0.15} |
> | carry_stuff_hugegravity | {’torso’: 4.0, ‘thigh’: 1.0, ‘shin’: 1.0, ‘foot’: 1.0, ‘gravity’: 1.5} |
> | tinyfoot_moon | {’foot’: 0.5, ‘gravity’: 0.15} |
>
> **SSDE Network Configurations:**
>
> - **Policy Network:** Input(18) → fc(1024) → fc(1024) → fc(1024) → fc(1024) → output(6)
> - **Critic Network:** Input(18) → fc(256) → fc(256) → fc(256) → fc(256) → output(1)
>
> **Hyperparameter Configurations:**
>
> | **SAC Hyperparameters** |  |
> | --- | --- |
> | Lr for Policy | 3e-4 |
> | Lr for Critic | 3e-4 |
> | Reward Scaling | 10 |
> | Target Output Std | 0.1 |
> | Policy Update Delay | 4 |
> | Target Update Delay | 4 |
> | Batch Size | 256 |
> | Replay Buffer Size | 1e6 |
> | **SSDE Hyperparameters** |  |
> | Sparsity Ratio $\lambda_{\Gamma}$ | 1e-3 |
> | Sparsity Ratio $\lambda_{\Lambda}$ | 1e-3 |
> | Trade-off Parameter $\beta$ | 0.6 |
> | Dormant Threshold $\tau$ | 0.6 |
> | Dormant Reset Interval | 3e4 |
>
> We are deeply grateful for the opportunity for continual engagement with the reviewer. If there are any additional concerns or suggestions, we would be happy to further discuss and address them. Thank you once again for your time and thoughtful input.
>
> [1] Gaya, Jean-Baptiste, et al. "Building a subspace of policies for scalable continual learning." ICLR 2023.
>
> [2] Sun, et al. "Rewiring neurons in non-stationary environments." NeurIPS 2023.

---

> ### Author Response · Authors · 2024-12-01
> **Gentle Reminder**
>
> Dear Reviewer q2Ui,
>
> I hope this message finds you well. As the discussion deadline of **December 2nd (AoE)** approaches, we wanted to follow up regarding the responses we provided to your comments and concerns. Your feedback has been invaluable in helping us refine our work, and we hope our explanations have addressed your queries effectively.
>
> If there are any aspects that remain unclear or require further clarification, please do not hesitate to let us know. We are fully committed to providing additional details to ensure all your concerns are thoroughly addressed.
>
> Once again, we sincerely appreciate your thoughtful and meticulous review. Your insights have significantly enhanced the rigor and completeness of our work, and we are deeply grateful for your contributions.
>
> Best regards,
>
> Authors

---

### Official Review · Reviewer_f4aE · 2024-11-03

**Soundness:** 2
**Presentation:** 2
**Contribution:** 2
**Rating:** 5
**Confidence:** 4

**Summary:**

This paper proposes a structure-based method, SSDE, to enhance the plasticity-stability trade-off in continual reinforcement learning. It introduces a fine-grained allocation strategy that decomposes network parameters into fixed forward-transfer and task-specific trainable components. Furthermore, to improve the model’s exploration capability, this paper presents an exploration technique based on sensitivity-guided dormant neurons. Experiments conducted on the Continual World benchmark demonstrate that the proposed method achieves a superior success rate and outperforms current state-of-the-art methods in the CW10 task.

**Strengths:**

This paper addresses a critical issue in continual reinforcement learning: balancing plasticity and stability to mitigate catastrophic forgetting. The proposed method offers greater computational efficiency than existing approaches. Experimental results are promising, demonstrating that the proposed method achieves a higher success rate and outperforms other baseline methods on the CW10-v1 Continual World benchmark.

**Weaknesses:**

1. The paper can be improved in terms of writing/presentation.
2. From Table 3, it is evident that the author’s proposed method generally performs weaker than the ClonEx-SAC method. Therefore, will the ClonEx-SAC method continue to outperform the author’s method as the number of sequential tasks increases? Additionally, why doesn’t the forward transfer ability, or generalization ability, of the author’s method improve as the number of tasks increases?
3. The proposed sensitivity-guided dormant neurons offer limited novelty.
4. The author does not include comparisons with the latest regularization-based methods in continuous reinforcement learning, and the multi-task methods referenced are also outdated.

**Questions:**

1. If the differences between tasks are substantial, could using fixed forward-transfer parameters introduce issues that reduce flexibility?
2. Can the method proposed by the author continue adapting to additional tasks, and can its task completion performance still outperform other methods?
3. For additional issues, please refer to the weaknesses section.

---

> ### Author Response · Authors · 2024-11-20
> **Response to Reviewer f4aE (1/3)**
>
> We sincerely thank Reviewer f4aE for their thoughtful and constructive feedback. Please find our response (marked as **A**) to the reviewer comments below.
>
> > **[Weaknesses]**W**1: Writing/Presentation**
>
> **A**: We appreciate the reviewer’s feedback on writing and presentation. We have upload  the revised paper that improves the presentation and clarify technical explanations. We will further improve our writing and presentation in the future versions.
>
> > **[Weaknesses] W2(a): SSDE generally performs weaker than ClonEx-SAC.**
>
> **A:** Among the three major evaluation metrics **$P(\uparrow)$, $F(\downarrow)$** and **$FT(\uparrow)$,** SSDE performs weaker than ClonEx only on $FT(\uparrow)$**.** For the primary metric of task performance $P(\uparrow)$, SSDE outperforms ClonEx-SAC by 9% on CW10, and performs on par with ClonEx on CW20 (0.87 vs. 0.87). For $F(\downarrow)$, SSDE essentially outperforms ClonEx.
>
> It is important to highlight that **a higher $FT(\uparrow)$ doesn’t necessarily lead to higher $P(\uparrow)$**. This is because $FT(\uparrow)$ is an indicative measure over the progress of learning over the *area* of learning curve, while $P(\uparrow)$ measures the cumulative rewards at a fixed *point*. For instance, an algorithm that progresses faster due to extensive data rehearsal may not necessarily converge to a high cumulative reward.
>
> $FT(\uparrow)$ is inherently influenced by the level of access to prior data and models. Rehearsal-based methods like ClonEx require storing expert data or policy parameters from previous tasks, allowing them to train with significantly more information than structure-based methods like SSDE and CoTASP. This advantage enables faster learning and higher $FT(\uparrow)$, as the metric evaluates the AUC for the learning curve subtracted by a standard SAC baseline.
>
> Despite this, **SSDE** achieves a noticeably strong $FT(\uparrow)$ score without accessing any prior data or policies, leading to state-of-the-art $FT(\uparrow)$ score among structure-based methods and significantly outperforming its closest counterpart, CoTASP. Furthermore, since SSDE is rehearsal-free, it is also **more computationally efficient than ClonEx**. It is also worth noting that ClonEx is not open-sourced, and its paper lacks key details necessary for reproducibility, such as the amount of expert data stored for each seen task.
>
> We believe the algorithm design and empirical findings in SSDE will bring fresh insights to the community. Our proposed co-allocation strategy performs fine-grained allocation and inference for the forward-transfer and trainable task-specific parameters, fostering both structural plasticity and stability through its structure-based design. This unified framework enables SSDE to achieve a strong balance across metrics without relying on rehearsal or prior data storage, setting it apart from existing structure-based and rehearsal-based methods. Additionally, it also paves the way for more efficient and scalable solutions in continual RL.
>
> We also encourage the reviewer to refer to **[Weakness] W2 from Reviewer q2Ui** for additional related discussions.
>
> > **[Weaknesses]** **W2(b): Why doesn’t forward transfer, or generalization ability improve for SSDE as the number of tasks increase?**
>
> **A:** There is no inherent guarantee that $FT(\uparrow)$ score keep improve as more tasks are introduced. Instead, the increase in task diversity and complexity often hinders the forward transfer capabilities.
>
> For ClonEx, the $FT(\uparrow)$ score significantly improves on CW20 compared to CW10, **primarily  due to the repeating nature of the CW20 task sequence**. That is, after the competing the first 10 tasks, CW20 repeat the same 10 tasks again, thus making ClonEx gain access to expert data and policies for *all tasks in CW20*. This advantage effectively allows ClonEx to rehearse and fine-tune its policy across repeated tasks under the guidance of optimal teachers, explaining the noticeable increase in its $FT(\uparrow)$ score.
>
> In contrast, **SSDE treats each incoming task as a new task**, prompting a separate sub-network through its co-allocation strategy. SSDE’s task embedding-based sparse prompting not only enables the new task to inherit learned parameters from related previous tasks, but also assigns them dedicated trainable parameter capacity. This intuitive design ensures a balance between leveraging knowledge transfer and maintaining the flexibility to adapt to task-specific requirements.
>
> In most realistic scenarios, continual learning tasks are non-identical and diverse, making SSDE’s design more practical for real-world applications (e.g., once we know the task is identical to some previous ones, we can directly reuse the old policy instead of training it again). Our co-allocation strategy ensures that SSDE remains robust across heterogeneous task sequences.

---

> ### Author Response · Authors · 2024-11-20
> **Response to Reviewer f4aE (2/3)**
>
> > **[Weaknesses] W3: Sensitivity-guided dormant offers limited novelty.**
>
> **A:** Our work is not a straightforward application of dormant neurons proposed in [1] to continual RL domains.  It is important to emphasize that **our sensitivity-guided dormant score is fundamentally distinct from the original dormant score proposed in [1]**. Our new formulation is developed based on a key insight in continual RL: the failure of sparse sub-networks in handling challenging exploration tasks often stems from a loss of sensitivity to the input changes, a critical aspect that is not adequately captured by the original dormant neuron definition.
>
> We illustrate this motivation in detail through an intuitive case study on a hard exploration task, *stick-pull,* from continual-world (Appendix A.4.1). A sub-optimal policy fails to respond to the goal position, a crucial feature for guiding the robot to pull the stick successfully. The original dormant score proposed in [1], only examining the **scale of activation** for neurons, would fail to account for the input sensitivity of policy to input changes. We propose a novel dormant score function to bridges this gap by linking the observation distribution to neuron activation scores, fostering a more effective mechanism for identifying neurons that lack responsiveness to input changes, enabling the sparse sub-network to better adapt to the complexities of continual learning.
>
> Despite of the critical importance of input sensitivity in continual RL, most existing structure-based methods, such as PackNet and CoTASP, focus exclusively on “how to allocate” parameters neglecting “how to enhance the expressiveness of sparse policies” during training. This oversight limits the plasticity of the allocated sub-networks in handling complex and diverse tasks. We believe our work not only significantly extends the capabilities of structure-based continual RL policies but also shifts attention in the field toward a dual focus on “how to allocate” and “how to train with enhanced expressivity.”
>
> **Additional results**: We provide a comparison between our proposed sensitivity-guided dormant score and the original dormant score function from ReDo. Training the sparse sub-network using ReDo’s dormant score formula results in an average success rate that is **7% lower** than that achieved with SSDE in CW10. This highlights the superior effectiveness of our sensitivity-guided dormant score in enhancing the expressiveness of sparse sub-networks for structure-based continual RL. Extended discussion can be found in Appendix A.4.1.
>
> | Dormant Metric | Average Success |
> | --- | --- |
> | Redo | 0.88$\pm$0.02 |
> | Sensitivity | **0.95$\pm$0.02** |
>
> [1] Sokar, Ghada, et al. "The dormant neuron phenomenon in deep reinforcement learning." ICML 2023.

---

> ### Author Response · Authors · 2024-11-20
> **Response to Reviewer f4aE (3/3)**
>
> > **[Weaknesses]** **W4: Does not compare with latest regularization-based methods.**
>
> **A:** We did not include comparisons with regularization-based methods because, as established in the literature, **these methods generally struggle to handle complex continual RL tasks**, particularly those involving high task diversity and a long sequences like the CW10 and CW20 benchmarks.
>
> Instead, we focused on comparing SSDE with structure-based counterparts like CoTASP and PackNet, which align more closely with SSDE’s design and constraints, as well as the state-of-the-art rehearsal-based method ClonEx, which achieves strong performance.
>
> We are happy to extend the discussion or provide additional comparisons to regularization-based methods in the context of continual RL, should the reviewer have such works to suggest.
>
> > **[Question]** **Q1: If difference between tasks are substantial, could using fixed forward-transfer parameters introduce issues that reduce flexibility?**
>
> **A:** The success of structure-based methods in achieving a 0% forgetting score $F(\downarrow)$ relies on fixing the forward-transfer parameters. This approach ensures that knowledge transfer occurs exclusively in the forward path, where knowledge from previous tasks is transferred to subsequent tasks, while preventing backward transfer that could overwrite previously learned information.
>
> To address potential challenges arising from substantial task differences, our work incorporates several techniques designed to enhance the flexibility:
>
> - **Task Embedding-Based Sparse Prompting:** the number of forward-transfer parameters would be reduced and new tasks are ensured to be allocated with dedicated amount of trainable parameters to capture diverse tasks.
> - **Fine-Grained Inference Function (Eq. 6):**  our inference mechanism offers flexible control over the influence of previous tasks, mitigating the rigidity often associated with fixed forward-transfer parameters.
>
> Our work not only extends the capabilities of structure-based continual RL methods but also introduces fresh perspectives that inspire interesting follow-up research to tackle challenges associated with diverse and complex task sequences.
>
> > **[Question] Q2: Adapting to additional task**
>
> **A:** Yes. Our allocation strategy is **task-incremental,** making it flexible to accommodate new tasks. When a new task arrives, we could compute its task embedding, and apply the co-allocation strategy to generate a sub-network structure.
>
> However, the number of tasks that can be added is constrained by the total parameters of the model. It is foreseeable that as the number of tasks increases beyond a certain point, the model's performance will be affected. This phenomenon is universal, and many methods face a bottleneck in the number of tasks they can handle.
>
> In contrast, our method does not require prior knowledge of the total number of tasks, as is necessary information like PackNet, to determine the allocation strategy. This provides greater flexibility to some extent.
>
> We conducted additional experiments and deployed our method in the locomotion scenario Halfcheetah-compositional from Brax, as suggested by **Reviewer** **xXp4** The results listed below and detailed information can be found in Appendix A.4.4.
>
> |  | Performance | Model Size | Transfer | Forgetting |
> | --- | --- | --- | --- | --- |
> | CSP [1] | 0.69$\pm$0.09 | 3.4$\pm$1.5 | -0.31$\pm$0.09 | 0.0$\pm$0.0 |
> | Rewire [2] | 0.88$\pm$0.09 | 2.1$\pm$0.0 | -0.18$\pm$0.09 | -0.0$\pm$0.0 |
> | SSDE | **1.04$\pm$0.05** | 15.7$\pm$0.0 | 0.04$\pm$0.05 | 0.0$\pm$0.0 |
>
> [1] Gaya, Jean-Baptiste, et al. "Building a subspace of policies for scalable continual learning." ICLR 2023.
>
> [2] Sun, et al. "Rewiring neurons in non-stationary environments." NeurIPS 2023.

---

> ### Author Response · Authors · 2024-11-22
> **Gentle Reminder to Review Our Rebuttal**
>
> Dear Reviewer f4aE,
>
> We are sincerely grateful for your thoughtful and detailed feedback on our manuscript. Your comments, particularly regarding the comparison with ClonEx-SAC, are helpful in guiding us to improve our work.
>
> We have been working carefully to address your concerns, and we hope having a further discussion with you that would greatly help us improve the quality and clarity of our work.
>
> We understand how busy you must be, but we kindly wish to remind you of the upcoming discussion period deadline on November 26, 2024 (AoE). We deeply value your expertise and insights, which would greatly help us refine and enhance the quality of our manuscript.
>
> Thank you again for your time and support. It means a great deal to us.
>
> With gratitude,
>
> The Authors

---

> ### Author Response · Authors · 2024-11-24
> **Follow-Up on Rebuttal Feedback**
>
> Dear Reviewer f4aE,
>
> I hope this message finds you well.
>
> As the discussion deadline of **Nov 26 AoE** approaches, approximately **68 hours** remain. We noticed that we have not yet received your feedback on our rebuttal. Could you please take a moment to review our rebuttal and share your feedback within this time frame? Your insights are crucial for the final evaluation of our submission. We greatly appreciate your expertise and look forward to your valuable comments.
>
> Thank you very much for your attention to this matter.
>
> Best regards,
>
> The Authors

---

> ### Author Response · Authors · 2024-12-01
> **Gentle Reminder**
>
> Dear Reviewer f4aE,
>
> I hope this message finds you well. With the discussion deadline of **December 2nd (AoE)** approaching, we wanted to reach out to ensure you have all the information needed to finalize your feedback. Should you have any additional questions or concerns, we would be more than happy to provide further clarification or address any remaining points.
>
> We also wish to let you know that the other reviewers have shared their responses and engaged in further discussions with us. Your insights are equally vital to refining our work, and we deeply value the thoughtful effort you have invested in the review process.
>
> Thank you once again for your meaningful contributions, which have been instrumental in improving the quality, clarity, and impact of our research.
>
> Best regards,
>
> Authors

---

### Official Review · Reviewer_LLqA · 2024-11-04

**Soundness:** 2
**Presentation:** 2
**Contribution:** 2
**Rating:** 5
**Confidence:** 4

**Summary:**

This paper proposed a new structure-based method for plasticity and stability tradeoff  in continual RL. Specifically, the authors proposed (1) a sub-network co-allocation method which enhances meta sparse prompting by task-specific sparse prompting, (2) a finegrained masking method which only freeze exact parameters trained in previous tasks, and (3) a periodic resetting strategy for dormant neurons. However, this work is largely built upon previous works, which limits its technical contributions. And the reported experimental results are somewhat misleading. Thus, I lean towards reject at current stage.

**Strengths:**

1. The authors proposed to separate sub-network sparse prompting into global and task-specific levels, which seems effective in their ablation study.

2. The introduction of parameter-level masking and dormant neuron resetting techniques is beneficial for mitigating plasticity loss and keeping the learning capability of networks.

3. Combining the proposed techniques, the overall framework achieves higher computation efficiency.

**Weaknesses:**

1. This paper largely follows the technical parts of [1][2], which limits their own contribution. In detail,

1.1. The authors claim a crucial contribution of introducing task-level sparse prompting, which has also been proposed in [1], where the task dictionary is obtained through a dictionary learning algorithm using previous tasks' optimized prompts and their embedding. Here the authors sampled the task dictionary from N(0, 1) which shares the same distribution of global dictionary. I don't see the motivation of this choice. Could the authors elaborate more on this part and also detail the difference between [1] and their method?

1.2. In 4.2, since periodic dormant neuron resetting is well established in [2] for continual RL, it is inappropriate to claim this as a key contribution of their framework. In addition, the use of "exploration" is arguably misleading, which has various meanings in different contexts. I would suggest the authors use the specific term "structural exploration" throughout the paper for better clarification.

2. There are many model choices made without sufficient justification. In addtion to 1.1, the proposed "sensitivity dormant scores" also lacks clear motivation. What is the advantage of this metric than previously defined dormant scores? And ablation study is necessary for justifying such choices.

3. The experiments results seem misleading.

3.1. Throughout the experiments, their reproduced results of CoTASP has a huge gap with the reported ones in CoTASP's paper (e.g. for P: 0.73 v.s. 0.92 in CW10 and 0.74 v.s. 0.88 in CW20), which has no essential difference with SSDE (CW10) or even better (CW20). And the generation performance of SSDE is not reported for a fair comparison with previous methods.

3.2. In addition, the results are quite misleading given the reported P in CoTASP. In Table 4, the average success of SSDE w/o Dormant is 0.85 while that of CoTASP which also don't use Dormant neuron resetting achieves 0.92. Does this indicate that the proposed co-allocation strategy is inferior to the allocation method in [1] which also use sparse prompting?

4. The overall frameworks contain a lot of hyperparameters, such as the sparsity controlling parameters in Equation (3-4), the trade-off paramter in Equation (6), and the dormant threshold in Definition 4.1, which make the framework less practical. In addition, no ablation results are provided to show the robustness of the system to these hyperparameters.

[1] Yang, Yijun, et al. "Continual task allocation in meta-policy network via sparse prompting." International Conference on Machine Learning. PMLR, 2023.

[2] Sokar, Ghada, et al. "The dormant neuron phenomenon in deep reinforcement learning." International Conference on Machine Learning. PMLR, 2023.

**Questions:**

**Regarding Weakness 1**:

Q1.1. Could the authors explicitly compare their task-level sparse prompting approach to that in [1], highlighting key methodological differences, and discuss any potential advantages of their approach over the method in [1]?

Q1.2. Could the authors explain their motivation for sampling the task dictionary from N(0,1) rather than using dictionary learning?

Q1.3. Could the authors clarify how their use of dormant neuron resetting differs from or improves upon the approach in [2], and explain why they consider it a key contribution despite its prior use in continual RL?

**Regarding Weakness 2**:

Q2.1. Could the authors provide clear explanation of the advantages of their "sensitivity dormant scores" over previous dormant score definitions, and provide an ablation study comparing their "sensitivity dormant scores"  to other dormant score metrics?

Q2.2. Could the authors provide theoretical or empirical justification for their key model choices?

**Regarding Weakness 3.1**:

Q3.1. Could the authors explain the discrepancy between their reproduced CoTASP results and those reported in the original paper?

Q3.2. Could the authors provide generation performance results for SSDE for a fair comparison?

Q3.3. Could the authors discuss how the performance gap affects their conclusions about SSDE's effectiveness compared to CoTASP?

**Regarding Weakness 3.2**:

Q3.4. Could the author explain why their method performs inferior to [1] when dormant neuron resetting is not used, and provide additional analysis or experiments to clarify whether their co-allocation strategy offers advantages over the method in [1]?

Q3.5. Could the authors discuss potential reasons for the performance difference when dormant neuron resetting is not used and its implications for their method's effectiveness?

**Regarding Weakness 4**:

Q4.1. Could the authors provide a sensitivity analysis or ablation study for key hyperparameters to demonstrate the method's robustness?

Q4.2. Could the authors provide a discussion of strategies for hyperparameter selection in practical applications?

Q4.3. Could the authors provide a comparison of the number and sensitivity of hyperparameters in SSDE to those in baseline methods like CoTASP?

---

> ### Author Response · Authors · 2024-11-20
> **Response to Reviewer LLqA (1/5)**
>
> We sincerely thank the reviewer LLqA for their thoughtful and constructive feedback. Please find our response (marked as **A**) to the reviewer comments below.
>
> > **[Questions]  Q1.1 compare SSDE task-level sparse prompting approach to that in [1]**
>
> **A**: CoTASP focuses primarily on sub-network allocation. SSDE systematically extends this approach by addressing critical challenges in CoTASP for continual RL. Below, we provide a detailed comparison to highlight the novel aspects of SSDE:
>
> 1. **Single sparse prompting vs co-allocation.**
>
>     CoTASP employs a sparse coding-based allocation mechanism that assigns a single sparse prompting ($\alpha$) to each task based on task embeddings. This $\alpha$ activates specific neurons in the network layers to form sub-networks for tasks. During RL training, the sub-networks and $\alpha$ are optimized iteratively through dictionary learning and parameter updates.
>
>     However, CoTASP faces two major issues:
>
>     - Overlapping $\alpha$ allocations: Sub-networks for similar tasks may significantly overlap, leaving insufficient trainable parameters for latter tasks.
>     - Unstable training: The iterative updates to $\alpha$ mean that sub-network allocations are not fixed, which can lead to training instability.
>
>     SSDE introduces a novel co-allocation strategy (Section 4.1) to address these limitations. It combines two allocation approaches:
>
>     1. A fixed, shared task dictionary $D$, similar to CoTASP but without iterative updates.
>     2. A local, task-specific dictionary $D$, randomly mapping task embeddings to sub-networks.
>
>     This co-allocation strategy resolves the overlapping issue by ensuring a balance between plasticity and stability. Additionally, since sub-networks are preemptively allocated, SSDE eliminates the need for iterative updates during training, achieving higher computational efficiency.
>
> 2. **Neuron-level sub-network masking vs fine-grained sub-network masking.**
>
>     CoTASP activates sub-networks at the neuron level for task inference, freezing neurons used in previous tasks. However, this approach has two drawbacks:
>
>     - Parameters associated with inactive neurons in previous layers cannot be optimized, even if activated in subsequent layers.
>     - This inefficient parameter utilization limits the model’s overall adaptability.
>
>     SSDE overcomes these inefficiencies through a fine-grained masking mechanism (Section 4.2). Instead of freezing entire neurons, SSDE masks only the parameters actively used in the task sub-network. This ensures all parameters remain optimizable during continual RL, leading to **improved network utilization**. Additionally, SSDE incorporates a **trade-off parameter** (Eq. 6) that dynamically balances the contributions of forward-transfer (frozen) and task-specific (trainable) parameters. This mechanism improves performance by adapting the influence of previously learned knowledge during task inference.
>
> 3.  **Training with enhanced expressiveness**
>
>     CoTASP does not explicitly consider the expressiveness of the allocated sub-networks during training. As the number of trainable parameters decreases with each new task, CoTASP struggles to model complex behaviors effectively.
>
>     SSDE addresses this limitation by introducing **sensitivity-guided dormant neurons**, a mechanism designed to identify and reactivate underutilized neurons based on their input sensitivity. By reactivating these dormant neurons, SSDE ensures that sparse sub-networks retain sufficient representational power to tackle challenging and diverse tasks. This innovation overcomes the limitations inherent in CoTASP’s static allocation, enabling the model to maintain both adaptability and performance in continual RL scenarios.
>
>
> We empirically compare our co-allocation with CoTASP in experiment section 5.4, where the variant of our SSDE uses only co-allocation sparse prompting with the fine-grained masking but sensitivity-guided dormant neurons and $\beta$ mechanism are disabled. The $P(\uparrow)$ scores of all SSDE variants are presented in Table 4, Section 5.4 Ablation Study (in our new draft). All variants results are better than CoTASP’s score 0.73. They empirically verify the all components.
>
> In conclusion, these interconnected contributions significantly enhance the capability of SSDE. SSDE delivers improved stability, adaptability, and efficiency, as validated through superior performance and network utilization metrics. Our method not only advances the state-of-the-art performance for continual RL but also lays a solid foundation for future research in this area.
>
> [1] Yang, et al. "Continual task allocation in meta-policy network via sparse prompting." ICML 2023.

---

> ### Author Response · Authors · 2024-11-20
> **Response to Reviewer LLqA (2/5)**
>
> > **[Weaknesses] W1.1 and [Questions] Q1.2 : Motivation of sampling task dictionary from N(0,1).**
>
> **A**: The motivations of both task dictionary learning in CoTASP[1] and sampling task dictionary from N(0,1) in our SSDE are to facilitate the efficient network neuron allocation, i.e. higher network utilization / more trainable parameters. CoTASP[1] requires dictionary learning because their allocation mask, i.e. task embeddings $\alpha_t$, keeps changing in RL training. Such changing allocation mask slightly improve network utilization. We propose co-allocation approach in SSDE, which is sufficient to significantly improve network utilization. Figure 4 in experiment section supports that SSDE has **higher** **network utilization** and better performance compared to CoTASP. Therefore, SSDE can use sampling task dictionary from N(0,1), which reduce learning complexity in the algorithm.
>
> In CoTASP, the dictionary learning process adjusts the task dictionary after each task to mitigate excessive overlap between task embeddings as the number of tasks increase. **This overlap can severely constrain the capacity for trainable parameters in the allocated sub-network.** However, despite the additional computational overhead and optimization difficulty associated with dictionary learning, its direct contribution to increasing trainable parameter capacity or improving policy training outcomes remains **minimal**.
>
> SSDE adopts a more computationally efficient mechanism by sampling the task dictionary from N(0,1). Inspired by the well-established Locality Sensitive Hashing theorem [2], this method projects task embeddings onto a random plane in a learning-free manner. Sampling from the random task distribution effectively **address the overlap issue** by allowing us to generate alternative planes that allocate dedicated trainable parameter capacity without the need for iterative dictionary optimization. By removing the dictionary optimization, our allocation strategy becomes completely preemptive, significantly enhancing the allocation efficiency.
>
> Empirical results further validate the effectiveness of our proposed co-allocation strategy with task dictionary sampled from N(0,1). As shown in Figure 4(left), our proposed co-allocation strategy significantly enhances network utilization compared to CoTASP, highlighting its superior parameter allocation capabilities.
>
> > **[Weaknesses] W1.2 and [Questions] Q1.3: Periodic dormant neuron resetting is well established in [3] for continual RL.**
>
> **A:**  We would like to clarify that the ReDo mechanism introduced in [3] addresses a different conceptual framework of continual RL compared to the problem studied in our work.
>
> The continual RL problem in [3] focuses on policy updates in a single-task context, where targets (e.g., Q-values) change over time, but the task itself remains static. And the ReDo mechanism was developed to enhance the expressiveness of policy under such single-task learning scenarios. In contrast, the continual RL problem we address involves **task switches**, where the modeling of task relationships to handle task distribution shifts effectively is the key.
>
> While [3] highlights the dormant neuron phenomenon in deep RL, it does not provide a solution tailored to **task-switching continual RL scenarios with spare sub-policies**. Specifically:
>
> - In [3], the dormant neuron phenomenon is illustrated using statistical measures of the amount of dormant neurons on CIFAR-10, a standard (non-RL) continual learning task, to demonstrate the general existence of dormant neurons across machine learning problems. However, for policy training, ReDo is applied exclusively single-task Atari 100K training to improve policy expressiveness under static task conditions.
> - The ReDo mechanism itself does not include any task similarity modeling or multi-task handling.
>
> **Our contribution:**
>
> Our work extends the dormant neuron phenomenon to tackle an overlooked challenge of **structure-based continual RL**, where tasks solved using dedicated sparse sub-networks. These sub-networks inherently face expressiveness limitations due to their constrained parameter space, with a large portion of parameters frozen and only a small subset trainable for new tasks. Additionally, our approach is not a straightforward application of ReDo’s dormant score function to our problem. Instead, we propose **sensitivity-guided dormant neurons.**
>
> We provide a detailed case study in Appendix A.4.1 to illustrate this insight, highlighting the motivation and intuition.
>
>
> [1] Yang, et al. "Continual task allocation in meta-policy network via sparse prompting." ICML 2023.
>
> [2] Gionis, Aristides, Piotr Indyk, and Rajeev Motwani. "Similarity search in high dimensions via hashing." Vldb. Vol. 99. No. 6. 1999.
>
> [3] Sokar, Ghada, et al. "The dormant neuron phenomenon in deep reinforcement learning." ICML 2023.

---

> ### Author Response · Authors · 2024-11-20
> **Response to Reviewer LLqA (3/5)**
>
> > **[Questions] Q1.3: How their use of dormant neuron resetting differs from or improves upon the approach in [1].**
>
> **A:**  ReDo mechanism in [1] identifies dormant neurons based solely on their activation scale. In contrast, our sensitivity-guided dormant score incorporates the relationship between neuron activation and the input observation distribution. The new form of dormant score is grounded by a critical insight that structure-based continual RL policies often lose sensitivity to input variations, leading to suboptimal behavior.
>
> The new score function enables our method to identify neurons that fail to respond to input changes, which is crucial for addressing the expressiveness challenges of sparse sub-networks in structure-based continual RL. Additionally, [1] focuses on standard full-scale single-task policy, whereas our method is tailored to sparse sub-network policies for structure-based continual RL.
>
> Our method not only highlights a new expressivity challenge within the sparse policy networks for structure-based continual RL but also contributes a new dormant score function to advance the research community.
>
> > **[Questions] Q2.1: Advantage of “sensitivity dormant scores” vs previous defined dormant scores from [1].**
>
> **A:**  The ReDo mechanism from [1] measures neuron expressiveness solely by measuring on activation magnitude, an approach that works well for fully connected single-task policies but falls short in addressing the unique challenge of sparse sub-networks in structure-based continual RL, where limited trainable parameters and progressive freezing exacerbate expressiveness issues.
>
> **Our sensitivity-guided dormant score goes beyond measuring activation magnitude by linking neuron activations to input sensitivity.** This improvement allows us to identify neurons that fail to adapt to significant changes in the input distribution, which often hinders policy learning in sparse sub-networks, e.g., ignoring the change of goals in manipulation (Appendix A.4.1).
>
> By reactivating these underutilized neurons, our approach significantly enhances the adaptability and expressiveness of sparse sub-networks, making it better suited for complex continual RL tasks.
>
> > **[Questions] Q2.2: Provide ablation study comparing “sensitivity dormant scores” to other dormant score metrics.**
>
> **A:**  Thank you for suggesting a valid comparison. We have included additional experimental results in Table 4 comparing our sensitivity-guided dormant mechanism to the ReDo mechanism proposed in [1].  These results demonstrate the effectiveness of our sensitivity-guided dormant mechanism in addressing expressivity challenge of sparse sub-networks for structure-based continual RL.
>
> | Dormant Metrics | Average Success |
> | --- | --- |
> | ReDo [1] | 0.88$\pm$0.02 |
> | SSDE | **0.95**$\pm$**0.02** |
>
> [1] Sokar, Ghada, et al. "The dormant neuron phenomenon in deep reinforcement learning." ICML 2023.

---

> ### Author Response · Authors · 2024-11-20
> **Response to Reviewer LLqA (4/5)**
>
> > **[Questions] Q3.1 and Q3.3: Gap for reproduced vs. reported results for CoTASP.**
>
> **A:** We use reproduced results for CoTASP in our paper because we observed a notable gap between our reproduced scores and the reported scores from the original paper. We consulted the authors and found out that **they had developed modified reward functions for a subset of tasks in the continual world benchmark**, which contributed to their reported performance. Unfortunately, the exact changes to these altered reward functions were not tractable, making us impossible to replicate.
>
> To ensure a fair comparison, we compare with CoTASP under our reproduced performance scores, endorsed by the CoTASP authors. It is important to note that SSDE was **trained on the original environments from continual world benchmarks** without modifying the underlying reward functions or settings.
>
> We also respectfully highlight that SSDE achieves a performance higher than CoTASP’s *reported* score (0.92) on CW10, showcasing the strong capability of our approach.
>
> > **[Questions] Q3.2: Generation performance of SSDE is not reported.**
>
> **A:** We are slightly unclear on what is specifically meant by "generation performance" in this context. If it refers to the comparison experiments mentioned in **[Questions]  Q1.1** regarding the explicit comparison of training continual RL policy only with the task-level sparse prompting generated by SSDE to CoTASP, we have already provided the corresponding results in **Figure 4** and **Table 4** of our paper. However, if "generation performance" pertains to another aspect, we would be happy to clarify further upon receiving additional details or context.
>
> > **[Weaknesses] W3.2: Average success of SSDE w/o dormant is 0.85, does this indicate SSDE’s co-allocation strategy is inferior to CoTASP’s sparse prompting?**
>
> **A:** We respectfully disagree SSDE’s co-allocation strategy is inferior to CoTASP’s sparse prompting. The score of 0.85 for SSDE w/o dormant highlights that our proposed co-allocation strategy can generate **more effective** sub-network structure compared to CoTASP’s sparse prompting algorithm, which scores 0.73.
>
> The primary reason for SSDE’s superior allocation strategy over CoTASP is due to: (I) our co-allocation results in increased number of free (trainable) parameters to be updated for capturing new skills (II) the trade-off parameter $\beta$ employed in the fine-grained inference can achieve a better balance between forward transfer and task-specific update.
>
> We provide extended discussion on comparing co-allocation vs. CoTASP’s sparse prompting in Section 5.2 and Section 5.4, where we showcase comprehensive results regarding to learning curves, network utilization, as well allocation time.

---

> ### Author Response · Authors · 2024-11-20
> **Response to Reviewer LLqA (5/5)**
>
> > **[Questions] Q3.4 Could the author explain why their method performs inferior to [1] when dormant neuron resetting is not used, and provide additional analysis or experiments to clarify whether their co-allocation strategy offers advantages over the method in [1]?**
>
> **A:** When the dormant neuron mechanism is disabled, our method achieves an ablation result of $0.85 \pm 0.06$, which is significantly higher than the performance of [1] ($0.73 \pm 0.11$). This highlights the inherent strength of our co-allocation strategy even without the additional enhancement of dormant neuron resetting. Notably, our result is also comparable to ClonEx-SAC’s $0.86 \pm 0.02$, despite SSDE operating without access to prior task data or policies for replay. This reinforces the computational efficiency and robustness of our approach in comparison to methods that rely on data storage and rehearsal.
>
> > **[Questions] Q3.5 Could the authors discuss potential reasons for the performance difference when dormant neuron resetting is not used and its implications for their method's effectiveness?**
>
> **A:** Sparse prompting-based methods like SSDE and CoTASP allocate and train a sparse sub-network for each task, in contrast to approaches such as ClonEx and PackNet, which train dense networks at full capacity (PackNet then applies post-training pruning to make the policy sparse). Within the allocated sparse sub-network, a significant portion of parameters are frozen to preserve stability, making expressivity an inherent challenge. Dormant neuron resetting provides a practical and effective mechanism to enhance the expressivity of these sparse networks, enabling better adaptation to new tasks while maintaining stability.
>
> > **[Weaknesses] W4 and [Questions] Q4.1 and 4.3: Ablation results.**
>
> **A:** We have conducted a detailed sensitivity analysis for the key hyperparameters in our method.
>
> | Threshold | 0.2 | 0.4 | 0.6 | 0.8 |
> | --- | --- | --- | --- | --- |
> | average success | 0.86$\pm$0.01 | 0.88$\pm$0.02 | **0.95**$\pm$**0.02** | 0.83$\pm$0.06 |
>
> | beta | 0.1 | 0.2 | 0.3 | 0.4 | 0.5 | 0.6 | 0.7 | 0.8 | 0.9 | 1.0 |
> | --- | --- | --- | --- | --- | --- | --- | --- | --- | --- | --- |
> | average success | 0.72$\pm$0.01 | 0.83$\pm$0.02 | **0.95**$\pm$**0.02** | 0.89$\pm$0.03 | 0.84$\pm$0.05 | 0.84$\pm$0.07 | 0.82$\pm$0.08 | 0.83$\pm$0.08 | 0.82$\pm$0.10 | 0.83$\pm$0.14 |
>
> > **[Questions] Q4.2 strategies for hyperparameter selection**
>
> **A**: Tuning SSDE is relatively straightforward. For the SAC algorithm, we adopt the hyperparameters recommended in the original Continual World benchmark, focusing primarily on exploring the policy and critic architectures. SSDE inherits the same network architecture as CoTASP, ensuring consistency and proven effective. For sparse coding, we use the same sparsity ratio suggested in CoTASP, which has proven effective. Regarding dormant neuron settings, we explore the threshold ratio $\tau$ and the resetting period within a reasonable range close to the values used in ReDo. For the trade-off parameter, we perform a grid search within [0.1–1.0], which allows for robust performance optimization.
>
> We acknowledge the significant contributions from both CoTASP and ReDo in providing robust hyperparameters that have greatly facilitated our research. Similarly, we hope that SSDE can contribute meaningful insights and serve as a strong foundation for future work in the community.
>
> [1] Yang, et al. "Continual task allocation in meta-policy network via sparse prompting." ICML 2023.

---

> ### Author Response · Authors · 2024-11-22
> **Gentle Reminder to Review Our Rebuttal**
>
> Dear Reviewer LLqA,
>
> We sincerely appreciate the time and effort you have dedicated to reviewing our manuscript. Your thoughtful feedback, particularly on the comparison with CoTASP and ReDo and your questions regarding the reproducibility of CoTASP's experimental results, are helpful in guiding us to improve our work.
>
> We understand how busy you must be, but we kindly wish to remind you of the upcoming discussion period deadline on November 26, 2024 (AoE). Your expertise and insights are highly valued, and any further feedback you could provide would greatly help us refine and enhance the quality of our manuscript.
>
> Thank you once again for your thoughtful comments and generous support.
>
> Warm regards,
>
> The Authors

---

> ### Author Response · Authors · 2024-11-24
> **Follow-Up on Rebuttal Feedback**
>
> Dear Reviewer LLqA,
>
> I hope this message finds you well.
>
> As the discussion deadline of **Nov 26 AoE** approaches, approximately **68 hours** remain. We noticed that we have not yet received your feedback on our rebuttal. Could you please take a moment to review our rebuttal and share your feedback within this time frame? Your insights are crucial for the final evaluation of our submission. We greatly appreciate your expertise and look forward to your valuable comments.
>
> Thank you very much for your attention to this matter.
>
> Best regards,
>
> The Authors

---

> ### Comment · Reviewer_LLqA · 2024-11-26
>
> Thanks for the detailed replies, which address most of my questions. However, I still have concerns on the trustworthiness of the reproduced CoTASP results (0.73 v.s. 0.92). The claims made in the replies largely rely on whether the reproduced CoTASP results can reflect the true performance of CoTASP. Since this gap is huge, I would say I hold a very cautious attitude towards the corresponding claims. If the authors of CoTASP uses modified reward functions to achieve the their results, is it possible to reproduce some of them and see whether the proposed method can also benefit from these extras? And it sounds confusing to me that CoTASP used these reward functions to boost the performance from 0.73 to 0.92 but reported no further details in their paper? I currently keep my score, but if ACs and other reviewers believe the results are reliable, then my score would be increased to 6 ("marginally above the acceptance threshold")

---

> > ### Author Response · Authors · 2024-11-28
> > **Addressing Concerns on Result Reproduction**
> >
> > We deeply appreciate the time and effort the reviewer has dedicated to evaluating our work. We also thank the reviewer for raising this thoughtful and valid concern regarding the reproduced CoTASP results.
> >
> > We would like to clarify that all CoTASP scores are generated by running the original CoTASP code (https://github.com/stevenyangyj/CoTASP) with identical settings, using the officially released commit.
> >
> > To enhance transparency and clarify the CoTASP baseline scores, we re-run CoTASP and **make the records available under an anonymous wandb account for the reviewer to examine.  All hyperparameters for each seed are clearly visible in the link,** and the algorithm was tested with the original CW10 benchmark without modifying the reward function.
> >
> > (https://wandb.ai/iclr_2025_ssde_continual_rl-iclr/CoTASP_Testing/reports/ICLR2025_CoTASP_Reproduce--VmlldzoxMDM1MzAzNg?accessToken=22xe9avpmoynbchfcwyg4utxqsypxsre5r4yxamyfs3wcstsajc0ygjq0hzats3t).
> >
> > We fully acknowledge the significant contribution of this prior work to the research community. When reproducing its scores, we also observed a discrepancy in performance and reached out to the CoTASP authors for clarification. The CoTASP authors kindly reviewed our CW configuration and checkpoint of their code and confirmed its correctness. We truly appreciate their time and efforts dedicated to help with reproducing the scores. They commented that the observed difference might stem from the reward function used in their experiments. However, as the exact details of the **reward modification were unavailable,** they advised us to use the scores obtained under the original CW reward function, acknowledging our reproduced results as valid for comparison. There are also some de-anonymized communication logs can be provided documenting these discussions.
> >
> > In academy, it is a common and accepted practice to run open-sourced code and report the reproduced scores as the baseline performance when the original settings and details are unavailable. We humbly note that it would neither be feasible nor our obligation to independently recreate the unknown reward function used in the original CoTASP experiments. We believe it is **fairer and more meaningful** to compare both methods under the **standard CW environment with unmodified reward functions.** In our paper, we have clearly highlighted that the CoTASP scores reported are based on our reproduction using the original CW environment settings.
> >
> > **Our proposed SSDE was developed with strict adherence to fair evaluation settings**, ensuring unbiased and reliable comparisons with other approaches. We are deeply committed to fostering reproducibility and transparency in the research community. To this end, **we plan to release our code in the future and submit both the CoTASP and SSDE results to the Continual World benchmark** (https://sites.google.com/view/continualworld/cw-challenge) and engage with the CW and CoTASP authors as appropriate.  We believe this effort will help ensure transparency in comparisons while also raising awareness of sparse prompting-based methods in continual reinforcement learning.
> >
> > If you have any further concerns or suggestions, we would be more than happy to address them. We greatly value your continued engagement and thoughtful feedback.
> >
> > Best Regards,
> >
> > Authors

---

> ### Author Response · Authors · 2024-12-01
> **Gentle Reminder**
>
> Dear Reviewer LLqA,
>
> I hope this message finds you well. As the discussion deadline of **December 2nd (AoE)** approaches, we would like to kindly remind you that we are available to address any additional questions or concerns you might have. Please do not hesitate to reach out—we are fully committed to providing any further clarification needed.
>
> **In response to your valuable feedback, we have published the results of the CoTASP experiment replication to enhance transparency and provide additional context. We hope this supplementary material adequately addresses your queries and strengthens the clarity of our work.**
>
> We are sincerely grateful for your detailed review and the thoughtful observations you have shared regarding the CoTASP results. Your feedback has been pivotal in improving the depth and comprehensiveness of our research, as well as in solidifying its contributions to the field.
>
> Thank you again for your time and effort in reviewing our work.
>
> Best regards,
>
> Authors

---

### Author Response · Authors · 2024-11-20
**General Response to Reviewers and Revision Submitted.**

We sincerely thank the reviewers for their thoughtful and constructive feedback. We are encouraged by the positive reception of our work, with several reviewers acknowledging SSDE’s state-of-the-art stability and competitive plasticity (Reviewer q2Ui, Reviewer f4aE) and its computational efficiency (Reviewer f4aE, Reviewer xXp4). The novelty and significance of our technical contributions were also highlighted (Reviewer xXp4, Reviewer f4aE).

In light of these valuable comments, we have conducted additional experiments, provided clarifications, and **revised our manuscript**. Specifically, we have made the following major updates:

1. Sensitivity analysis of hyperparameters $\beta$, $\tau$ (Reviewer q2Ui, Reviewer xXp4).
2. Comparison of sensitivity-guided dormant score from SSDE with original dormant score in ReDo (Reviewer LLqA, Reviewer xXp4).
3. Comparison of our co-allocation strategy with the sparse coding from CoTASP (Reviewer LLqA, Reviewer xXp4).
4. Additional results on the Locomotion task, to showcase the generality of task embedding-based co-allocation strategy (Reviewer q2Ui, Reviewer xXp4).
5. Clarification of difference between SSDE and CoTASP, ClonEx-SAC, and dormant neurons (Reviewer LLqA, Reviewer f4aE, Reviewer xXp4).

All changes have been highlighted in blue in the manuscripts.

**We want to restate the contribution of our paper to the research community:**

- Advancing sparse prompting-based method for continual RL, with efficiency Features:
    - No storage of previous tasks’ transitions or optimal policy.
    - No rehearsal or behavior cloning on previous data.
    - No prompt optimization or dictionary learning.
    - No pruning of the allocated sub-network structure.

    These features collectively address critical challenges in scalability, efficiency and computational overhead.

- Open-source continual RL solution for CW10 and CW20:
    - Achieved state-of-the-art performance for CW10 of 95%
    - Open-source implementation with high reproducibility and easy to extend for follow-up works
    - Results are robustly verified across multiple random seeds, ensuring reliability.
- Fine-grained sparse prompting method:
    - Introduced a co-allocation strategy that attends to allocation of forward-transfer/trainable parameters.
    - Proposed a fine-grained inference with trade-off parameters.
    - Formulated a novel inference function (Eq 6) for structure-based continual RL .
- Enhancing structural exploration with sensitivity-guided dormant neurons:
    - Highlighted the importance of input sensitivity for sparse prompting-based algorithms.
    - Proposed a new dormant score evaluation metric.

If any of the reviewers have any further questions, we would be pleased to answer them.

Best Regards,

Authors

---

### Author Response · Authors · 2024-12-04
**Summary of Reviewer Feedback and Rebuttal Response**

Dear Reviewers and Area Chair,

We sincerely thank all reviewers for their thoughtful feedback and invaluable insights, which have significantly enhanced the quality and clarity of our work. Our contributions have received notable recognition, including:

- **Effectiveness of Subnetwork Allocation**: *Reviewer LLqA and Reviewer xXp4* commended the design and performance of our subnetwork allocation mechanism.
- **Balancing Plasticity and Stability**: *Reviewer f4aE and Reviewer q2Ui* emphasized our method’s ability to effectively balance plasticity and stability, addressing catastrophic forgetting while maintaining computational efficiency.
- **Outstanding Benchmark Performance**: *All reviewers* unanimously acknowledged the outstanding performance of our approach, achieving state-of-the-art results on the CW10 benchmark.

In response to the key concerns raised during the review process, we have carefully addressed the following points in our rebuttal:

- Conducting a **sensitivity analysis** of the hyperparameters $\beta$ and $\tau$, emphasizing their critical roles in our model. (Reviewer q2Ui, Reviewer xXp4)
- Comparing the **sensitivity-guided dormant score** with the original dormant score from ReDo, demonstrating the superiority of our proposed metric. (Reviewer LLqA, Reviewer xXp4)
- Evaluating our **co-allocation strategy** against the sparse coding approach in CoTASP, highlighting the enhanced model utilization achieved by our method.  (Reviewer LLqA, Reviewer xXp4).
- Extending our experiments to **locomotion scenarios**, showcasing the versatility and adaptability of our approach across diverse applications. (Reviewer q2Ui, Reviewer xXp4)
- Addressing concerns regarding the **replication score** of CoTASP by providing our reproduced learning curves publicly on WANDB. (Reviewer LLqA)
- Further exploring the **impact and role of model size** in our method. (Reviewer q2Ui)

We are pleased that numerous reviewers recognized our efforts, noting that their concerns were thoroughly addressed and expressing gratitude for our detailed responses. We believe this article will help address the issue of continual learning, draw attention to sparse prompting related method, and our dormant metric will further deepen the understanding of dormancy of neurons. At the same time, as a reliable work, it will effectively advance the progress of the Continual World benchmark.

Thank you once again for your support and consideration.

Best regards,

The Authors

---

### Meta-Review · Area_Chair_QspE · 2024-12-20

**Metareview:**

To reduce catastrophic forgetting, which is a problem in continuous reinforcement learning, the paper proposes a new method called SSDE. SSDE divides the parameters of a task into fixed and trainable parameters and performs efficient co-allocation. It also uses inductive sensitivity-guided dormant neurons to reset insensitive parameters, improving flexibility in response to new tasks.

The strengths of this paper are The authors address the important problem of reducing catastrophic forgetting in continuous reinforcement learning by balancing plasticity and stability. The authors propose to separate sparse prompting of subnetworks into global and task-specific levels. The introduction of parameter-level masking and dormant neuron resetting techniques helps to reduce plasticity loss and preserve the network's learning ability.

The weaknesses of this work are as follows. There are problems with the way the paper is written and it is difficult for the reader to properly understand the content, especially the impact of this paper. There is insufficient discussion of the scope and limitations of the proposed method.

The final rating of this paper was three negative and one positive. The authors also responded to the reviewers' concerns, but were unable to fully resolve the reviewers' concerns. At this stage, it did not meet the ICLR acceptance threshold, but the AE recommends that the authors submit it to the next conference after considering the reviewers' comments.

**Additional Comments On Reviewer Discussion:**

The authors have made the following major updates. 1. Sensitivity analysis of the hyperparameters, 2. Comparison of the sensitivity-guided dormant scores from SSDE and the original dormant scores from ReDo, 3. Comparison of the authors' co-allocation strategy and CoTASP's sparse coding. 4. Additional results on locomotion tasks to demonstrate the generality of the task embedding based co-allocation strategy, 5. Clarification of the differences between SSDE, CoTASP, ClonEx-SAC and dormant neurons.

One reviewer commented that they had concerns about the reliability of the reproduced CoTASP results. The other reviewers who gave negative ratings did not respond. One reviewer has decided to accept the paper as all concerns have been addressed. The paper does not appear to have any technical problems, but the impact of the paper is not clearly communicated in its current form. It is a borderline paper, but at present it is difficult to find any strong elements that would push it over the threshold.

---

### Decision · Program_Chairs · 2025-01-22

Reject